# Warhead biosynthesis and the origin of structural diversity in hydroxamate metalloproteinase inhibitors

Franziska Leipoldt[1,2], Javier Santos-Aberturas[3], Dennis P. Stegmann[1,2], Felix Wolf[1,2], Andreas Kulik[4], Rodney Lacret [3], Désirée Popadić[3,5], Daniela Keinhörster[1], Norbert Kirchner[1,2], Paulina Bekiesch [1,6], Harald Gross [1,2], Andrew W. Truman [3] & Leonard Kaysser [1,2]

Metalloproteinase inhibitors often feature hydroxamate moieties to facilitate the chelation of metal ions in the catalytic center of target enzymes. Actinonin and matlystatins are potent metalloproteinase inhibitors that comprise rare *N*-hydroxy-2-pentyl-succinamic acid warheads. Here we report the identification and characterization of their biosynthetic pathways. By gene cluster comparison and a combination of precursor feeding studies, heterologous pathway expression and gene deletion experiments we are able to show that the *N*-hydroxy-alkyl-succinamic acid warhead is generated by an unprecedented variation of the ethylmalonyl-CoA pathway. Moreover, we present evidence that the remarkable structural diversity of matlystatin congeners originates from the activity of a decarboxylase-dehydrogenase enzyme with high similarity to enzymes that form epoxyketones. We further exploit this mechanism to direct the biosynthesis of non-natural matlystatin derivatives. Our work paves the way for follow-up studies on these fascinating pathways and allows the identification of new protease inhibitors by genome mining.

[1] Pharmaceutical Biology, Eberhard Karls University Tübingen, 72076 Tübingen, Germany. [2] German Center for Infection Research (DZIF), Partner site Tübingen, 72076 Tübingen, Germany. [3] Department of Molecular Microbiology, John Innes Centre, Colney Lane, Norwich NR4 7UH, UK. [4] Interfaculty Institute of Microbiology and Infection Medicine, Microbiology/Biotechnology, Eberhard Karls University Tübingen, 72076 Tübingen, Germany. [5] Present address: Institute of Pharmaceutical Sciences, University of Freiburg, Albertstr. 25, D-79104 Freiburg i. Br., Germany. [6] Present address: Department of Pharmacognosy, University of Vienna, Althanstrasse 14, A-1090 Vienna, Austria. Franziska Leipoldt and Javier Santos-Aberturas contributed equally to this work. Correspondence and requests for materials should be addressed to A.W.T. (email: andrew.truman@jic.ac.uk) or to L.K. (email: leonard.kaysser@pharm.uni-tuebingen.de)

nhibitors of proteases and protease-like enzymes have versatile applications in medicine and other areas, and are used in the clinic for the treatment of cancer, hypertension, thrombosis, diabetes as well as viral and bacterial infections. Most of these drugs are produced synthetically but a substantial part of them have been developed from or are inspired by natural products[1]. A frequent structural feature of protease inhibitors from nature is the presence of warheads that mediate their specific binding to the active site of target enzymes and may allow mechanism-based inhibition. Warheads often consist of electrophilic groups, which are prone to be attacked by nucleophilic residues generating covalent adducts. Such moieties include β-lactones, Michael systems, epoxyketones, and others[2].

Recently, we explored the biosynthetic mechanisms for the production of the α,β-epoxyketone warhead in the proteasome inhibitors epoxomicin and eponemycin (Fig. 1). We found that the peptidic backbone of the compounds is assembled by a hybrid nonribosomal peptide synthetase/polyketide synthase (NRPS/PKS)[3]. Further studies revealed that a conserved acyl-CoA dehydrogenase (ACAD, EpnF in the eponemycin and EpxF in the epoxomicin gene cluster) is essential for the conversion of a dimethyl-β-keto acid derivative into the bioactive α-methyl-α,β-epoxyketone[4,5]. In vitro studies of the epoxyketone synthase EpnF by Challis and co-workers showed that the reaction proceeds via an unprecedented decarboxylation–dehydrogenation–monooxygenation mechanism[6]. We used EpnF as a probe for genome mining and found a number of orphan pathways in various bacteria[3]. One of these gene clusters in *Actinomadura atramentaria* DSM 43919 sparked our interest because it encoded an unusual non-linear NRPS/PKS assembly line in combination with a putative ethylmalonyl-CoA (EMC) pathway. Here, we report that this orphan gene cluster in *A. atramentaria* DSM 43919 directs the biosynthesis of a family of structurally complex hydroxamate metalloproteinase inhibitors, the matlystatins. We are able to show that an ACAD enzyme, homologous to the epoxyketone synthase, is responsible for the generation of a diverse array of matlystatin congeners, which led us to direct the biosynthesis of non-natural matlystatin derivatives. The identification and analysis of the actinonin gene cluster, another metalloproteinase inhibitor, allowed us to postulate a unique biosynthetic route to the hydroxamate warhead of this class of molecules, which was supported by stable isotope feeding studies.

## Results

### Identification of the matlystatin biosynthetic gene cluster.

A protein homology search using EpnF, the epoxyketone synthase from the eponemycin biosynthetic gene cluster (BGC), led us to identify an unusual orphan pathway in *A. atramentaria* DSM 43919 (Fig. 2). The potential gene cluster contains 18 open reading frames (ORFs), 6 of which encode a putative hybrid NRPS/PKS assembly line. The presence of a single putative PKS module (MatO) and a thioesterase (TE, MatP) together with the ACAD homolog MatG suggested the production of an epoxyketone proteasome inhibitor. The domains for two putative NRPS modules are encoded in the cluster on five discrete genes (*matH, matI, matJ, matK, matO*). The non-linear organization of the NRPS genes, however, made it difficult to predict the composition of the produced peptide. A literature search revealed that another *A. atramentaria* isolate (SANK 61488) produces matlystatins[7]. The core structure of the matlystatins is a pseudotripeptide that consists of *N*-hydroxy-2-pentyl-succinamic acid, piperazic acid (Pip), and isoleucine[8]. The succinamic acid derivative serves as the warhead of the molecule in analogy to the identical moiety of actinonin. Hydroxamate warheads are common features of metalloproteinase inhibitors that facilitate chelation of the active site metal ion. Actinonin is a hydroxamate-containing antibiotic isolated from *Streptomyces* sp. in 1962 (Fig. 1)[9]. Based on its potent activity against the bacterial peptide deformylase, actinonin has served as a lead compound for the development of new antimicrobial agents, such as GSK1322322, which is currently evaluated in clinical phase II studies[10]. Its characteristic *N*-hydroxy-2-alkyl-succinamic acid warhead is shared by only a few other bacterial compounds: propioxatin, BE16627B, YM-24074, and the matlystatins (Fig. 1)[8,11–13]. However, despite their interesting biological activity and potential medicinal value, the biosynthesis of this class of hydroxamate protease inhibitors has so far not been investigated.

*A. atramentaria* SANK 61488 produces five different matlystatin congeners: A (**1**), B (**2**), D (**3**), E (**4**), and F (**5**) (Fig. 1), which differ in their C-terminal substitutions and the length of the fatty acid side chain. Matlystatin A is a potent inhibitor of MMP-2, -3, and -9 and has thus been considered as a lead compound for the development of anticancer drugs[14,15]. Reanalysis of the putative "epoxyketone" gene cluster from *A. atramentaria* DSM 43919 indicated that the cluster might indeed direct the biosynthesis of matlystatins. The adenylation

**Fig. 1** Chemical structures of selected protease inhibitors. Matlystatin A (**1**), matlystatin B (**2**), matlystatin D (**3**), matlystatin E (**4**), and matlystatin F (**5**)

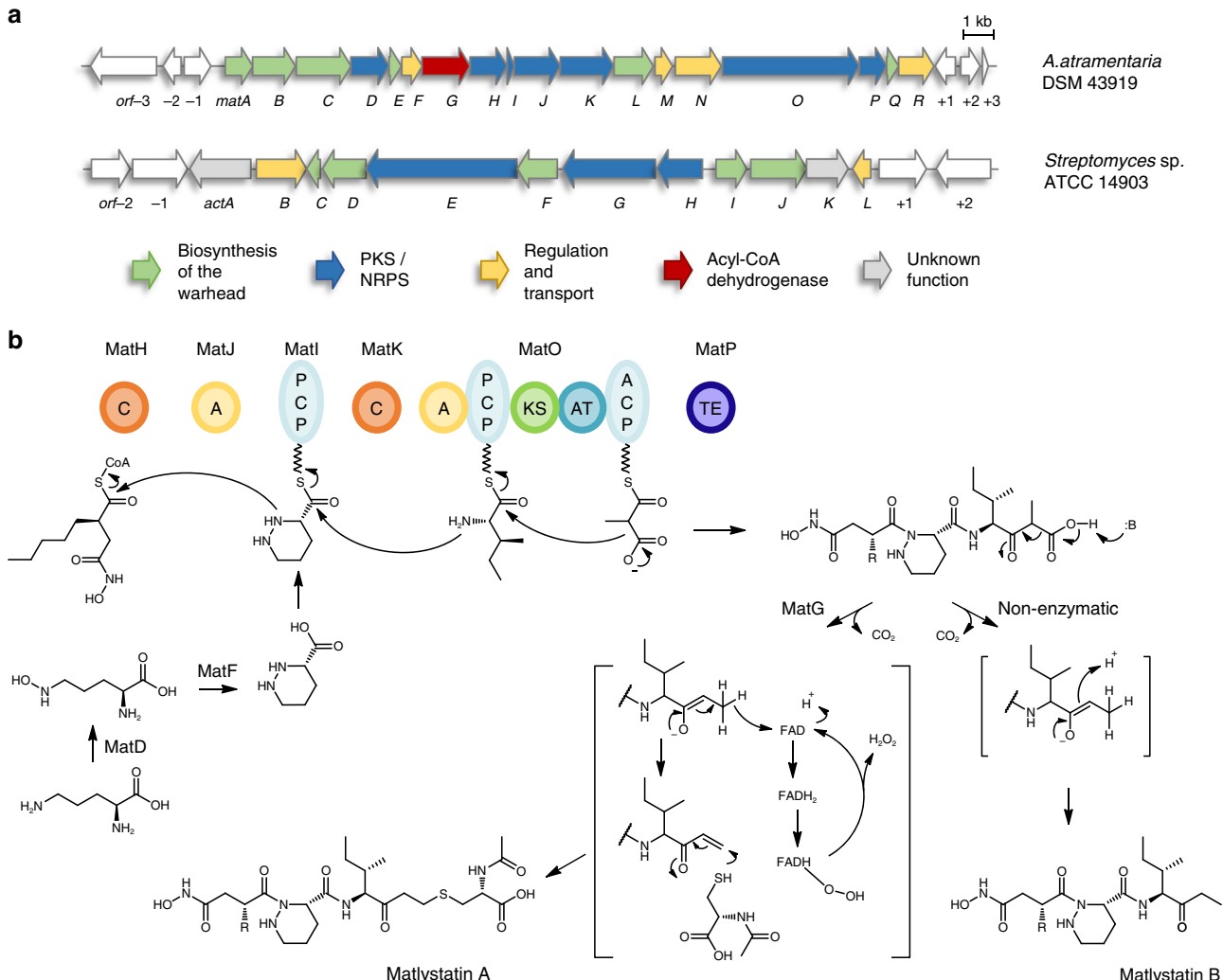

**Fig. 2** Biosynthesis of hydroxamate metalloproteinase inhibitors. **a** Organization of the *mat* and the *act* biosynthetic gene cluster from *A. atramentaria* DSM 43919 and *Streptomyces* sp. ATCC 14903 (NCIMB 8845), respectively. **b** Biosynthetic model for the assembly and modification of the matlystatins. A adenylation domain, ACP acyl carrier protein, AT acyl-transferase domain, C condensation domain, KS ketosynthase, PCP peptidyl carrier protein, TE thioesterase

(A)-domains in the gene cluster, MatJ and MatO, were predicted to activate Pip and leucine (Supplementary Table 3), respectively, with MatJ exhibiting 59% sequence identity to the marformycin Pip adenylation protein MfnK[16]. MatD and MatF have 48% and 49% sequence identity to the ornithine $N^5$-hydroxylase KtzI and the recently characterized heme-dependent enzyme KtzT from *Kutzneria* sp. 744, respectively. This enzyme pair has been shown to be responsible for the formation of the N–N bond of Pip via the generation of an *N*-hydroxylated intermediate[17,18]. We speculated that another *N*-oxygenase, MatA, could be responsible for the formation of the hydroxamate group. In addition, the gene cluster encodes proteins that could participate in an EMC-like pathway. The EMC pathway functions in many α-proteobacteria and actinomycetes for acetyl-CoA assimilation. Key reactions include the reductive α-carboxylation of crotonyl-CoA by a crotonyl-CoA carboxylase-reductase (CCR) to generate EMC and the subsequent 1,2-rearrangement by the ethylmalonyl-CoA mutase (ECM) to afford methylsuccinyl-CoA[19,20]. Given that the matlystatin warhead features a substituted succinyl moiety, we hypothesized that homologs of these proteins encoded in the *mat* gene cluster (MatL and MatBQ) might play a role in the synthesis of an alkylated succinic acid precursor.

To determine whether *A. atramentaria* DSM 43919 actually produces matlystatins, we subjected culture extracts of the strain to liquid chromatography-mass spectrometry (LC-MS) analysis and searched for the respective masses. The presence of matlystatins A, D/F, and E was readily detected, as were the deshydroxymatlystatins A (**1a**), B (**2a**), D/F (**3a/5a**), and E (**4a**) (Fig. 3a). The MS[2] fragmentation patterns of the matlystatins were indicative and matched the data from the literature (Supplementary Figs. 1–3)[8]. It was not possible to distinguish between the D and F isomers with LC-MS[2], but peak splitting in the MS chromatogram with maxima at 16.6 and 17.1 min suggests the formation of both congeners (Supplementary Fig. 4). In order to evaluate the production rates of the deshydroxymatlystatin derivatives, we isolated **1a**, **2a**, and **3a/5a** from cultures of *A. atramentaria* DSM 43919, yielding 33, 3, and 8 mg/L, respectively.

**Heterologous expression of the *mat* gene cluster**. Having established that *A. atramentaria* DSM 43919 is indeed a producer of matlystatins, we wanted to confirm that these molecules are produced by the identified orphan pathway. Therefore, we

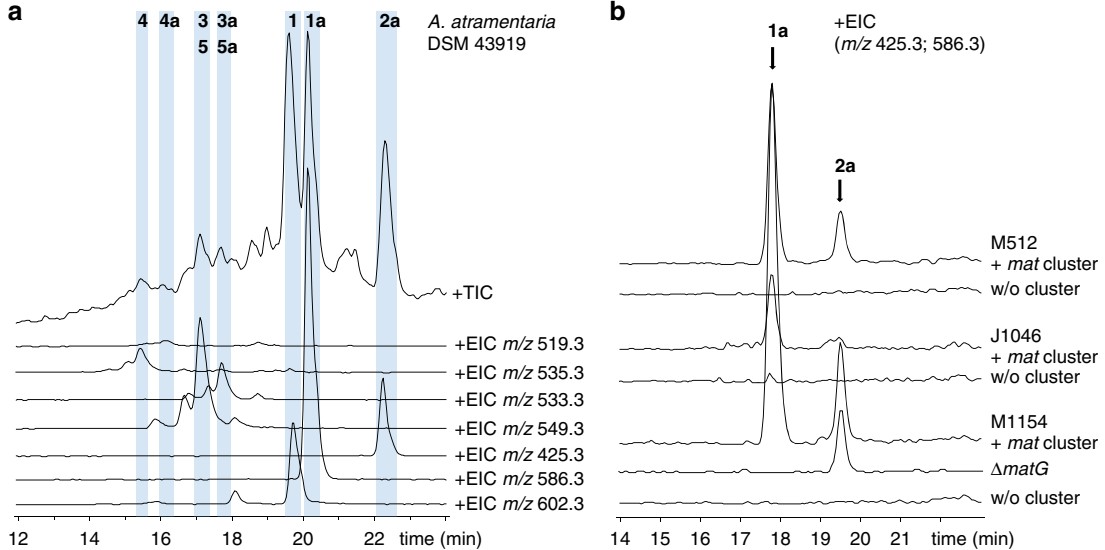

**Fig. 3** LC-MS analysis for the detection of matlystatin derivatives in culture extracts. **a** Matlystatin production in extracts of *A. atramentaria* DSM 43919. **b** Heterologous production of deshydroxymatlystatins in *Streptomyces* species: *S. coelicolor* M512 (M512), *S. albus* J1046 (J1046), and *S. coelicolor* M1154 (M1154). All presented phenotypes of the mutants were reproducible in at least two other clones and one repeated independent experiment

constructed a pCC1FOS-based genomic library of *A. atramentaria* DSM 43919 and found fosmid 7C11 to contain the complete putative gene cluster. We then generated fosmid matDK01 by replacing the chloramphenicol resistance gene in the backbone of 7C11 with a restriction cassette that contained elements for site-specific integration into *Streptomyces* chromosomes[21,22]. To improve the prospect of a successful heterologous expression, we selected different strains as possible surrogates: *Streptomyces coelicolor* M512, *S. coelicolor* M1154, and *S. albus* J1074. Butanol culture extracts of strains with and without the cluster were analyzed by LC-MS and MS[2] and compared to the wild-type producer. The accumulation of deshydroxy derivatives of matlystatin A (**1a**) and B (**2a**) was observed for all heterologous strains carrying the cluster (Fig. 3b). In addition, we found low amounts of deshydroxymatlystatins D/F only in extracts of the *S. coelicolor* M1154 derivatives with the cluster (Supplementary Fig. 5). For the following studies, we thus used *S. coelicolor* M1154 as a host. This result clearly confirmed that the identified orphan pathway, which was previously presumed to produce an epoxyketone proteasome inhibitor, in fact directs the biosynthesis of the metalloproteinase inhibitors matlystatins. We were able to recover deshydroxymatlystatin A from culture extracts of the heterologous producer in yields of 16 mg/L. However, hydroxylated matlystatins could only be found in trace amounts. This is consistent with the small proportion of these molecules produced by the wild-type strain. The instability of the hydroxamate warhead in matlystatins has been reported before and might vary in different cultivation media. Another explanation could be that the function of MatA is impaired in the heterologous host.

**Deletion of the putative ACAD gene *matG*.** Based on our findings, we were intrigued by the role of the EpnF homolog MatG in matlystatin biosynthesis. First, we wanted to determine if any matlystatin derivatives are produced by the cluster that contain an epoxyketone warhead but had previously been overlooked. However, we were not able to find such molecules in culture extracts of either the wild-type or the heterologous producer using LC-MS. To acquire information on the relevance of *matG* for matlystatin biosynthesis, we generated an in-frame gene deletion mutant in *S. coelicolor* M1154 using the PCR targeting method[23]. Interestingly, LC-MS analysis of culture extracts from

the Δ*matG* mutant showed that production of **1a** was entirely abolished, whereas **2a** was still accumulated (Fig. 3b). On the basis of these results, we speculated that MatG participates in the modification of the C-terminus of the peptide backbone. For the epoxyketone synthase EpnF, Challis and co-workers recently showed that it uses an α-dimethylated β-keto acid as a substrate[6]. This precursor derives from an NRPS peptidic product in which the terminal leucine condenses with a malonyl unit, which is then α-methylated twice from *S*-adenosylmethionine by the action of a single PKS module containing a methyltransferase domain. The authors postulated for EpnF that after initial decarboxylation to generate an enolate intermediate, hydride is eliminated from the β-carbon atom to afford an α,β-unsaturated ketone, thereby reducing the flavin cofactor. In the last step, the reduced flavin would react with molecular oxygen to provide a hydroperoxide species which is able to transform the unsaturated ketone to the epoxyketone[6]. By analogy, the EpnF homolog MatG might also facilitate the generation of an unsaturated ketone intermediate via decarboxylation and hydride transfer of a β-keto acid substrate (Fig. 2b). The produced vinyl ketone is a strong electrophile and might, in the absence of a flavin hydroperoxide, be attacked by nucleophiles such as the sulfhydryl group of *N*-acetylcysteine or a secondary amine. Instead of an epoxide group as in the epoxyketones, MatG would therefore assist in forming carbon–sulfur or carbon–nitrogen bonds as found in the matlystatins A and D–F. This is consistent with the abolishment of the production of these molecules following deletion of *matG*, whereas matlystatin B could still be produced via spontaneous decarboxylation of the β-keto acid precursor.

**Feeding of [13]C-labeled propionate and L-ornithine.** Our hypothesis on the mechanism of MatG would require that the enzyme acts on a α-monomethylated β-keto acid substrate. Because the putative PKS module on MatO does not include a methyltransferase domain, the α-methyl group would likely derive from the incorporation of methylmalonyl-CoA. In order to explore the origin of the carbons at the C-terminus of the peptide, we performed feeding experiments with [2-13C]-propionate in the matlystatin producer strain *A. atramentaria*. Culture extracts were analyzed by LC-MS and compared to cultures without supplementation (Fig. 4; Supplementary Fig. 6).

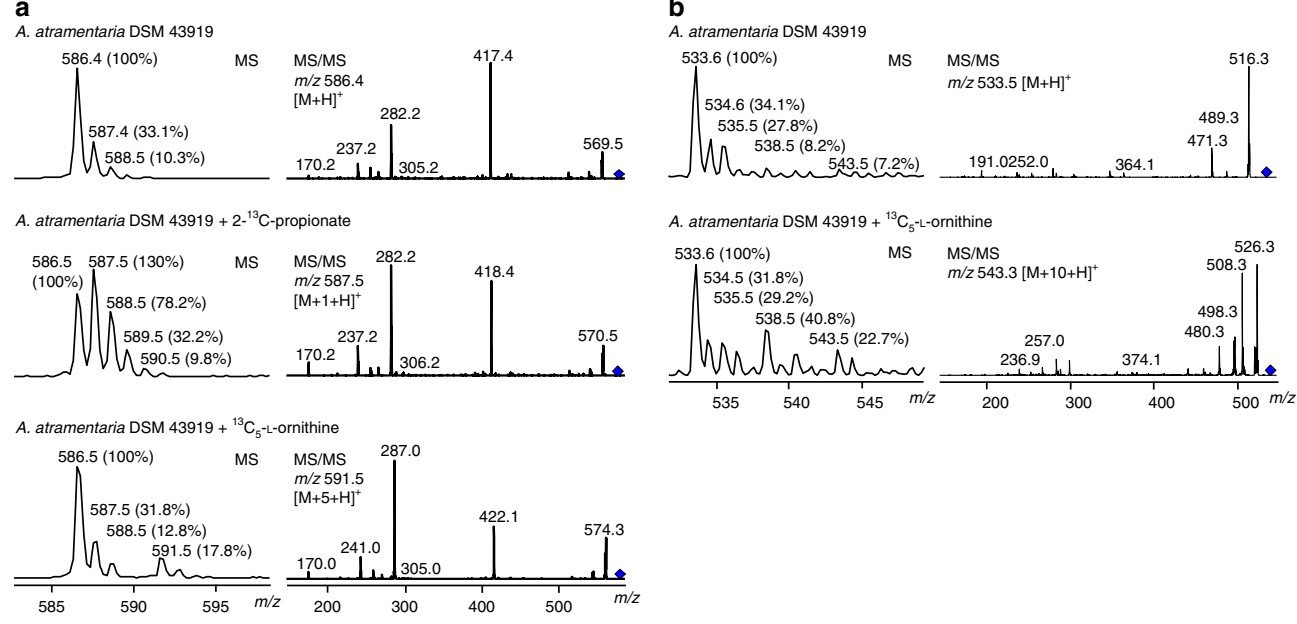

**Fig. 4** MS analysis of extracts from cultures of *A. atramentaria* DMS 43919 with and without supplementation with isotope-labeled precursors. Isotopic distribution and MS$^2$ spectrum of (**a**) **1a** (*m/z* 586.4 [M + H]$^+$) and (**b**) deshydroxymatlystatin D/F (*m/z* 533.6 [M + H]$^+$). Diamonds indicate the precursor ion. All presented chemotypes were reproducible in at least one other culture and one repeated independent experiment

In the cultures containing [2-$^{13}$C]-propionate, isotopic mass peaks *m/z* 587.3 [M + 1 + H]$^+$ for **1a**, *m/z* 426.3 [M + 1 + H]$^+$ for **2a**, and *m/z* 534.3 [M + 1 + H]$^+$ for the deshydroxymatlystatin D/F were substantially increased. These results indicate the incorporation of propionate in the matlystatins. The MS$^2$ spectrum of **1a** [M + 1 + H]$^+$ revealed fragment signals *m/z* 282.2 and 170.1, which are also found in the MS$^2$ spectra of the unlabeled compound (Fig. 4). However, the presence of fragment ions *m/z* 418.4 and 306.2 indicates incorporation of a $^{13}$C-labeled carbon at the C-terminus of the molecule. Similar observations were made in the MS$^2$ spectrum of **2a** (Supplementary Fig. 6). To unequivocally determine the position of the incorporated carbon from [2-$^{13}$C]-propionate in the matlystatins, we isolated the $^{13}$C-enriched deshydroxymatlystatin A (**1a**) from culture extracts in yields of 18.3 mg/L and analyzed it in comparison with its non-labeled form, employing $^{13}$C NMR spectroscopy. Due to a limited solubility of **1a** in $d_4$-MeOH and signal overlap of the enhanced resonance in $d_6$-DMSO, the measurement was conducted in $d_7$-DMF.

A complete assignment of **1a** in $d_7$-DMF was determined by using a routine set of 2D NMR spectra (Supplementary Table 5; Supplementary Figs. 7–12). Notably, in $d_7$-DMF, many resonances in the $^1$H and $^{13}$C NMR spectra of **1a** were doubled or broadened, indicating the presence of two conformers in a ratio of 3:2. This observation may be explained by the presence of *cis/trans* rotamers of the amide bond connecting the Pip and the isoleucine moiety. Ultimately, precursor incorporation studies using [2-$^{13}$C]-propionate showed a 14-fold $^{13}$C-enrichment at carbon C-2" (Supplementary Table 6; Supplementary Fig. 13).

One of the most intriguing questions in matlystatin biosynthesis is the origin of the unique annulated pyrazole heterocycle in the D/F derivatives (Fig. 1). Based on our previous results, we speculated that this ring system might derive from the reaction of the unusual amino acid Pip with a MatG product. Walsh and co-workers have shown that the Pip moiety in the kutznerides is generated from ornithine via an $N^5$-hydroxy-L-ornithine intermediate[17]. To test our hypothesis, we fed cultures of *A. atramentaria* with [U-$^{13}$C$_5$]-L-ornithine and compared the

extracts to those of unfed cultures. LC-MS analysis showed the incorporation of five labeled carbons in **1a** and **2a** with the increase of isotopic mass peaks of *m/z* 591.5 [M + 5 + H]$^+$ and 430.5 [M + 5 + H]$^+$, respectively (Fig. 4; Supplementary Fig. 6). Interestingly, the deshydroxymatlystatins D/F isotopic distribution revealed two new signals with *m/z* 538.5 [M + 5 + H]$^+$ and 543.5 [M + 10 + H]$^+$. This indicates that all matlystatins incorporate one molecule of ornithine, presumably as a Pip residue, into their backbone. In addition, the D/F derivatives contain a second ornithine-derived structural moiety. Analysis of the respective MS$^2$ spectra showed indicative fragment ions with *m/z* 526.3 (*m/z* 516.3 + 10), 374.1 (*m/z* 364.1 + 10), and 257.0 (*m/z* 252.0 + 5) for the incorporation of $^{13}$C$_5$ at the C-terminus of the molecules. This strongly suggests that the annulated pyrazole heterocycle in matlystatin D and F is synthesized from ornithine, and results from the condensation of Pip with an electrophilic precursor. Compared to matlystatin A, further oxidation is required at the C-terminus to generate the observed pyrazole, which could indicate an additional role for MatG or the action of another enzyme.

**MS networking and generation of unnatural matlystatins**. *A. atramentaria* was fermented in a variety of media in an attempt to optimize matlystatin production levels. Two media (TSB and bottromycin production medium, BPM) were identified that provided comparable levels of production to matlystatin production medium (MPM). Intriguingly, a preliminary LC-MS$^2$ analysis of these fermentations indicated the presence of unknown matlystatin congeners based on similar MS$^2$ patterns, especially the *m/z* 282.18 fragment that is consistent with fragmentation between Pip and isoleucine. We therefore assessed the combined MPM/BPM/TSB metabolomic data using mass spectral networking (Global Natural Products Social Molecular Networking[24,25]. http://gnps.ucsd.edu). This provided a detailed map of additional matlystatin congeners that either differ at the C-terminus of the molecule or feature a carboxylic acid instead of an amide or hydroxamic acid (Fig. 5a; Supplementary Figs. 14–16;

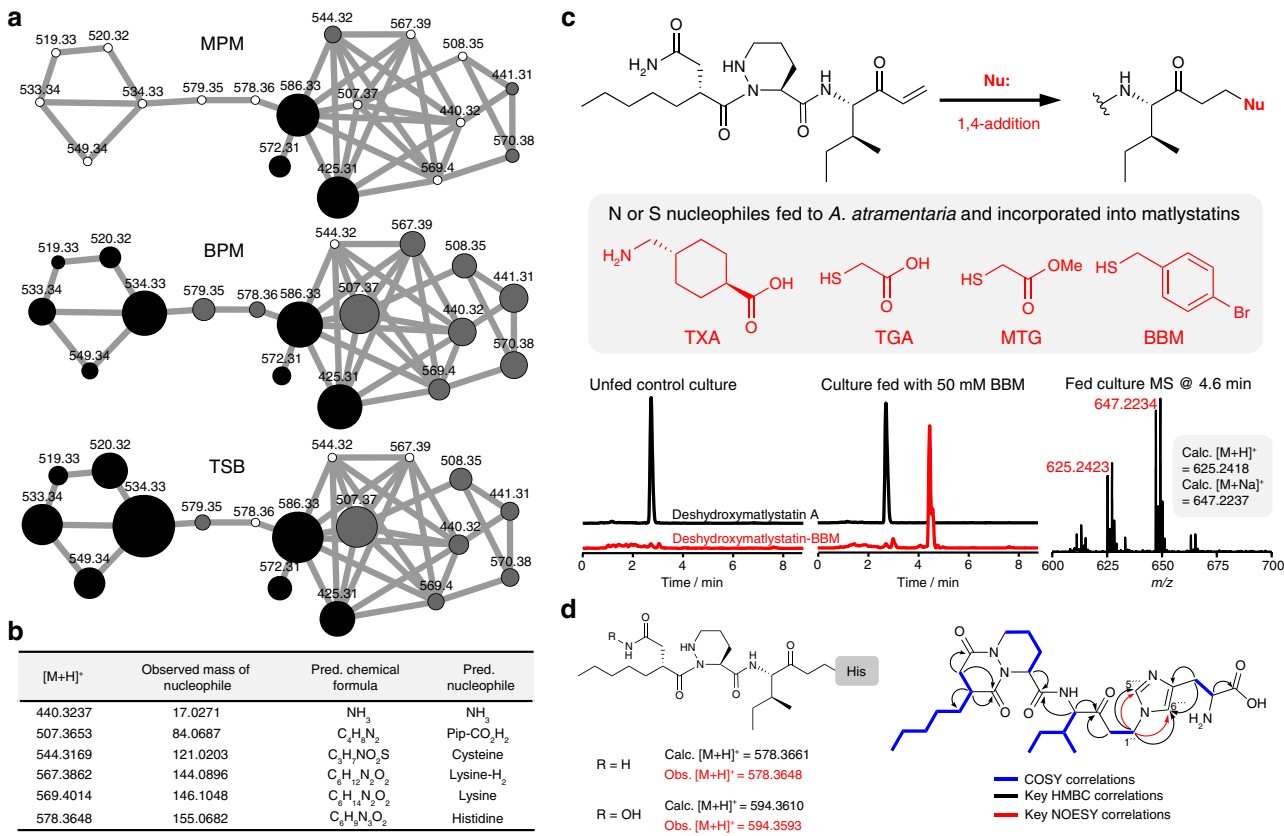

**Fig. 5** Production of matlystatin congeners. **a** MS networking analysis of matlystatin congeners produced by *A. atramentaria* DSM 43919 in different media. Black nodes indicate metabolites related (hydroxamate, amide, or carboxylate versions) to known matlystatin congeners, gray nodes indicate putative novel matlystatin congeners, and white nodes indicate congeners not detected in that medium. Nodes are sized to reflect the relative MS peak intensities. The corresponding cytoscape file of the network can be found in the Supplementary Data 1 (matlystatin_network.cys). **b** Predicted nucleophiles that generate novel deshydroxymatlystatin congeners. **c** Production of novel matlystatin congeners by feeding unnatural nucleophiles to *A. atramentaria* DSM 43919. LC-MS data for BBM feeding are shown. **d** Histidine-containing matlystatin congeners produced by fermentation in histidine-supplemented MPM

Supplementary Data 1). The carboxylic acid derivatives produced a characteristic MS² fragment of *m/z* 283.17. The relationship between these compounds and the known matlystatins was validated by detailed MSⁿ analysis (Supplementary Figs. 17–22), and the exact mass of these novel congeners was used to predict the likely nucleophiles that undergo 1,4-addition to the matlystatin vinyl ketone precursor (Fig. 5b). This suggested that various amino acids (or their derivatives) can react with this electrophilic intermediate. MS-based quantification of the new and known matlystatin congeners (Fig. 5a) indicated that their relative amounts vary considerably depending on the medium used. The diversity of compounds observed supports a model where nucleophiles react with the electrophilic vinyl ketone (Fig. 5c), either spontaneously or MatG-catalyzed, and the relative proportion of the products is influenced by the availability of nucleophiles in the production medium used.

To test this hypothesis, *A. atramentaria* was fermented in MPM supplemented with histidine, which yielded a significant amount of both the putative matlystatin-His and deshydroxymatlystatin-His congeners (*m/z* 594.36 and 578.36, respectively; Fig. 5d; Supplementary Fig. 23), as well as another related compound with *m/z* 561.34 (**6**). Crucially, without supplementation of histidine, these compounds were not detected in MPM culture extracts (Fig. 5a; Supplementary Fig. 23). **6** proved most amenable to purification from a large-scale fermentation at yields of 0.13 mg/L and was structurally characterized by MS² and NMR (¹H, ¹³C, COSY, HSQC, HMBC, NOESY; Fig. 5d; Supplementary Figs. 24–30; Supplementary

Table 7). This revealed a matlystatin-like compound with a histidine residue attached at the C-terminus and loss of hydroxylamine to yield a fused bicyclic ring. Crucially, HMBC and NOESY correlations between C1'' (4.25 ppm) and C5''' and C6''' on the histidine moiety determined that Nτ of histidine reacts with the vinyl ketone indicating that the feeding of unnatural nucleophiles could be used to direct the production of novel unnatural matlystatins. Therefore, we fed tranexamic acid (TXA), thioglycolic acid (TGA), methyl thioglycolate (MTG) and 4-bromobenzyl mercaptan (BBM) to separate cultures of *A. atramentaria*. This resulted in the production of compounds with *m/z* 580.4072 (calc. [M + H]⁺ = 580.4069), 537.2711 (calc. [M + Na]⁺ = 537.2717), 551.2878 (calc. [M + Na]⁺ = 551.2874), and 625.2423 (calc. [M + H]⁺ = 625.2418), respectively, that were not present in unfed controls (Fig. 5c; Supplementary Figs. 31–34). As with the natural congeners, MSⁿ analysis provided fragments of *m/z* 282.18 and 170.12 that are indicative of the conserved N-terminus of the deshydroxymatlystatins. Furthermore, the product of BBM feeding provided a characteristic isotopic pattern of a bromine-containing molecule (Fig. 5c).

**Gene deletion studies and cluster boundaries.** To gain further information on the genetic basis for matlystatin biosynthesis and to determine the cluster boundaries, we generated a set of additional gene deletion mutants. We separately substituted *orf-3* (putative cytochrome P450), *orf-2* (transcriptional regulator), *orf-1* (putative alcohol dehydrogenase), *matR* (transporter), *orf+1*

(transcriptional regulator), *orf+2* (putative 3-ketoacyl-ACP reductase), and *orf+3* (putative 4-oxalocrotonate tautomerase) with an apramycin resistance gene on fosmid matDK01. We also deleted the putative NRPS genes *matH* (C-domain), *matJ*

(A-domain), and *matK* (C-A-domain), and candidate genes for the formation of the warhead including the potential EMC pathway: *matB* (mutase), *matE* (epimerase), *matL* (crotonyl-CoA carboxylase/reductase), and *matQ* (mutase). After heterologous

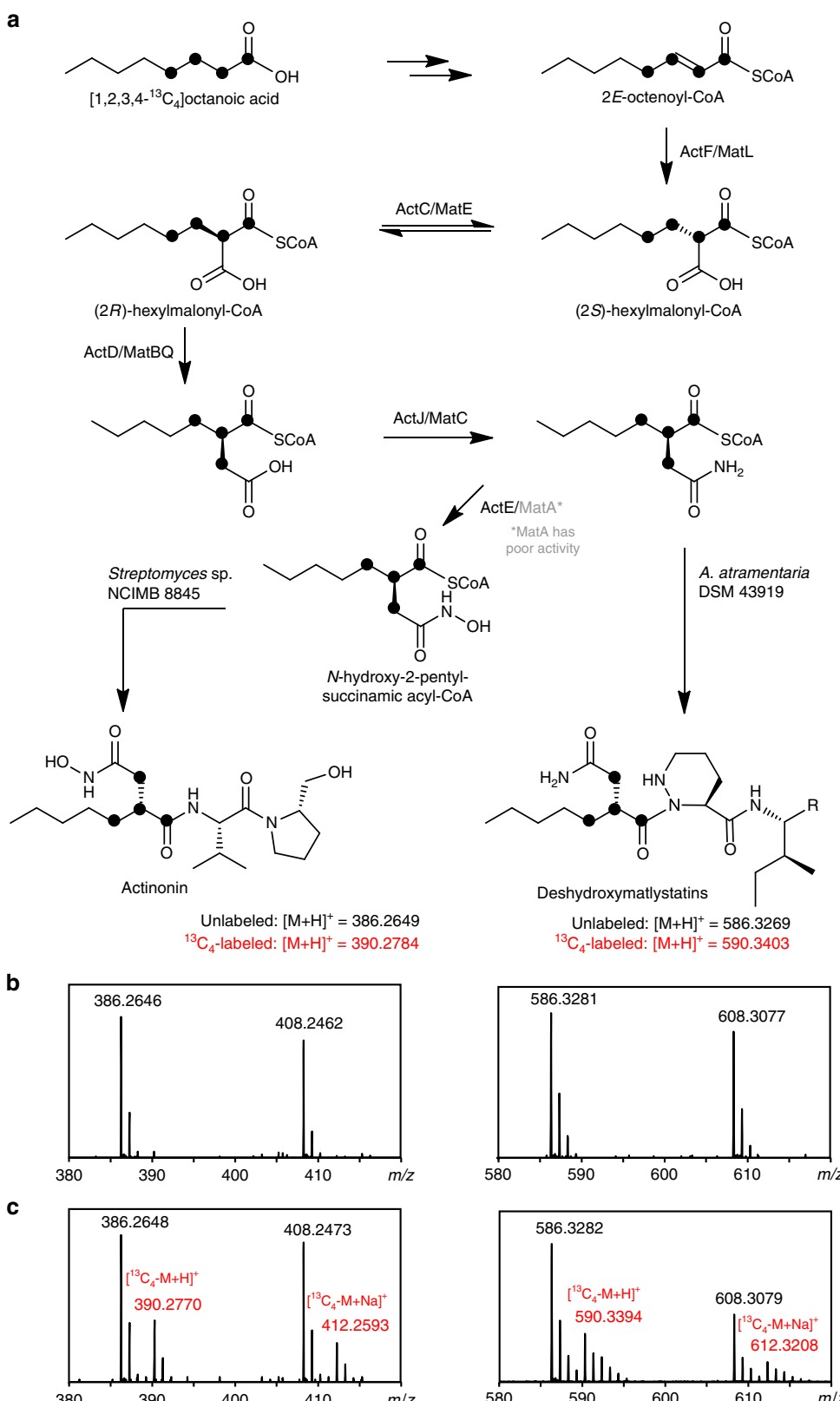

**Fig. 6** Model for the biosynthesis of the actinonin and matlystatin hydroxamate warhead. **a** Pathway and proposed incorporation of [1,2,3,4-$^{13}C_4$]-octanoic acid. **b** MS spectra of compounds from unlabeled media and **c** media fed with [1,2,3,4-$^{13}C_4$]-octanoic acid

expression of the mutant fosmids in *S. coelicolor* M1154, culture extracts were analyzed by LC-MS. Matlystatin formation was detected in the mutants Δ*orf-3*, Δ*orf-2*, Δ*orf*-1, Δ*matR*, Δ*orf+1*, Δ*orf+2*, and Δ*orf+3*. In contrast, deletion of *matA, matB, matC, matE, matH, matJ, matK, matL*, and *matQ* led to abolishment of matlystatin production (Supplementary Fig. 35). Taken together, this data indicate that the *mat* BGC consists of 17 putative genes, *matA* to *matQ*, spanning 24.1 kb. A gene table with the annotation results for the complete *mat* cluster can be found in the Supplementary Information (Supplementary Table 1). Our results from the gene deletion experiments strongly suggest that the putative EMC pathway encoded in *matB, matE, matL*, and *matQ* is essential for matlystatin biosynthesis. The EMC pathway commonly includes the reductive α-carboxylation of crotonyl-CoA by a CCR to generate EMC, and the subsequent 1,2-rearrangement by an EMC mutase to afford methylsuccinyl-CoA[19,20]. It is conceivable that in matlystatin biosynthesis a similar reaction sequence using 2-octenoyl-CoA as a precursor might lead to the intermediate 2-pentylsuccinyl-CoA. Such a pathway would be highly intriguing because the role of CCRs in secondary metabolism has so far been limited to the production of substituted malonyl-CoA derivatives, which are employed as non-canonical extender units in some PKS systems, while reports of ECM-like mutases have been restricted to methylmalonyl-CoA biosynthesis[26]. Notably, MatL is 66% identical to CinF, a 2*E*-octenoyl-CoA carboxylase from the cinnabaramide pathway[27]. The involvement of a dedicated EMC-like pathway in the generation of a natural product molecule has, to the best of our knowledge, not been reported.

**Identification of the actinonin biosynthetic gene cluster.** In order to evaluate our model on the biosynthesis of the hydroxamate warhead, we searched for other pathways of this class of metalloproteinase inhibitors. The most prominent example is actinonin, which contains the same *N*-hydroxy-2-pentylsuccinamic moiety as the matlystatins. Actinonin is reported to be produced by *Streptomyces* sp. ATCC 14903 (NCIMB 8845) and we validated actinonin production by comparison with a commercial standard. Genome sequencing of this strain resulted in a ~8.46 Mb draft sequence containing 31 putative pathways for the production of secondary metabolites. One of these gene clusters encodes a putative NRPS system in which the terminal module (ActE) comprises an A-domain with predicted specificity for proline (Supplementary Table 3) and a rare terminal reductase (Red) domain. In lyngbyatoxin and myxochelin biosynthesis, terminal Red domains have been shown to catalyze the four-electron reduction and release of peptidyl carrier protein (PCP)-bound intermediates as primary alcohols[28,29]. Such a biosynthetic mechanism would be perfectly suited to generate the proline-derived hydroxymethyl pyrrolidinyl moiety of actinonin and strongly indicates that the identified gene cluster represents the actinonin biosynthetic pathway. The *act* gene cluster contains a putative NRPS C-A-PCP module encoded by *actG*, and a discrete A-domain encoded by *actH*. Whereas the substrate specificity of the ActG A-domain could not be determined by in silico analysis, ActH is predicted to activate threonine (Supplementary Table 3). We propose that ActG, alone or in combination with ActH, activates and transfers valine to the ActG PCP. The ActG C-domain is proposed to catalyze the condensation of 2-pentylsuccinyl hydroxylaminyl-CoA with this valine residue. Subsequently, ActE would catalyze the condensation of the pseudodipeptide with PCP_ActE-bound proline, followed by reductive release of the resulting pseudotripeptide.

Importantly, the *act* gene cluster encodes the complete pathway for the predicted formation of the hydroxamate warhead

analogous to matlystatin biosynthesis: a CCR (ActF; 67% sequence identity to MatL), an EMC epimerase (ECE, ActC; 67% sequence identity to MatE), an ECM (ActD; 70% sequence identity to MatB), an asparagine synthetase (AsnB)-like enzyme (ActJ; 58% sequence identity to MatC), and a putative *N*-oxygenase (ActI; 48% sequence identity to MatA). ActD belongs to the 2-hydroxyisobutyryl-CoA mutase subclass, which function as heterodimers of a large mutase and a small cobalamin binding protein[30]. The matlystatin gene cluster encodes both (MatB and MatQ), whereas the actinonin gene cluster only appear to encode the large protein, although a protein with 71% identity to MatQ is found elsewhere in the *Streptomyces* sp. ATCC 14903 (NCIMB 8845) genome. The identification of an EMC-like pathway in both the matlystatin and actinonin BGCs provides strong support for a conserved biosynthetic route to the 2-pentylsuccinyl hydroxylaminyl moiety. A gene table with the annotation results for the complete *act* cluster can be found in the Supplementary Information (Supplementary Table 2).

**A biosynthetic model supported by $^{13}C_4$-octanoic acid feeding.** Based on the conserved series of genes across the *mat* and *act* gene clusters, we postulate the following reaction sequence for the generation of the distinct hydroxamate warhead in actinonin, matlystatin, and other metalloproteinase inhibitors of the same class (Fig. 6a). In the first step, the putative CCR (ActF/MatL) stereoselectively carboxylates octenoyl-CoA at the α-position as described for several homologs from both primary and secondary metabolism[19,31,32]. Next, an epimerase (ActC/MatE) interconverts (2*S*)-hexylmalonyl-CoA to the *R* epimer, which is the stereochemistry that is utilized by characterized mutases. A mutase (ActD/MatB) then catalyzes the adenosylcobalamin-dependent conversion of (2*R*)-hexylmalonyl-CoA to (2*R*)-2-pentylsuccinyl-CoA, a reaction that has not been described for previously reported mutases[20]. Subsequently, the free carboxyl moiety is amidated by an AsnB-like enzyme (ActJ/MatC), as in the biosynthesis of oxytetracyclines and sanglifehrin A[33,34]. The oxygenation of the amide is likely catalyzed by a putative *N*-oxygenase (ActI/MatA), a homolog to AurF from aureothin biosynthesis[35]. It has been shown that AurF is able to accumulate a hydroxylamine intermediate[36]. Alternatively, the enzyme pairs ActI/ActJ and MatA/MatC could act in a similar way to the recently characterized TsnB7 and TsnB9 in trichostatin biosynthesis. Here, the AurF-like oxygenase TsnB7 hydroxylates L-glutamine with the support of a helper protein TsnB6. Then L-glutamic acid γ-monohydroxamate serves as a donor for the transfer of a hydroxylamine group to the trichostatic acid precursor catalyzed by AsnB-like enzyme TsnB9[37]. However, a TsnB6 homolog could not be found in either the *act* or *mat* gene clusters, and the sequence identity of TsnB7 to ActI and MatA is rather low (37% and 34%, respectively). We would thus consider it more likely that hydroxylation of the amide occurs with (2*R*)-2-pentylsuccinamyl-CoA as a substrate. The reaction product (2*R*)-2-pentylsuccinyl hydroxylaminyl-CoA would then be used as an unusual acyl-CoA starter unit in the thiotemplate assembly lines of the actinonin and matlystatin pathways. The production of amide and carboxylic acid derivatives of matlystatin by *A. atramentaria* suggests that amidation and *N*-oxygenation may alternatively occur after peptide assembly.

To further test our biosynthetic proposal to the warhead, we used a stable isotope labeling strategy. Given that our proposed pathway starts with octenoyl-CoA, [1,2,3,4-$^{13}C_4$]-octanoic acid was fed to both producer strains with the expectation that it would be converted to the CoA thioester and oxidized in vivo to generate [1,2,3,4-$^{13}C_4$]-octenoyl-CoA. In *Streptomyces* sp. NCIMB 8845, this resulted in clear +4 peaks in comparison to

controls fed with unlabeled octanoic acid (Fig. 6b, c; Supplementary Fig. 36) for both the proton ($m/z$ 390.2770) and sodium ($m/z$ 412.2593) molecular ions. These adducts of actinonin could not be explained by degradation of octanoic acid and subsequent incorporation of $^{13}$C-labeled acetate. Furthermore, MS$^n$ analysis of the +4 peaks showed that all four $^{13}$C labels are incorporated into the 2-pentylsuccinyl hydroxylaminyl moiety portion of the molecule (Supplementary Fig. 37). An equivalent result was obtained for matlystatins when *A. atramentaria* was fed with [1,2,3,4-$^{13}$C$_4$]-octanoic acid. This was most clearly observed for deshydroxymatlystatin A (Fig. 6c), where a new peak of $m/z$ 590.3394 was observed by LC-MS. MS$^3$ experiments generated a fragment of $m/z$ 174.13 that is consistent with all labels being incorporated into the 2-pentylsuccinyl hydroxylaminyl moiety (vs. $m/z$ 170.12 for the unlabeled fragment; Supplementary Figs. 2, 38, 39). To further support these findings, we isolated $^{13}$C$_4$-actinonin from a [1,2,3,4-$^{13}$C$_4$]-octanoic acid fed culture for $^{13}$C NMR analysis in a yield of 0.42 mg/L. The cost of $^{13}$C$_4$-octanoic acid limited the quantity of $^{13}$C$_4$-actinonin that could be obtained, and therefore prevented full $^{13}$C NMR characterization. However, the $^{13}$C NMR spectrum featured four highly enriched signals: doublets at $\delta_C$ 33.5, 36.5, and 177.2, and a doublet of pseudo-triplets at $\delta_C$ 43.4 (Supplementary Fig. 40). A comparison with $^{13}$C and HSQC NMR spectra from an actinonin standard showed that these signals correspond to carbons 5, 2, 4, and 3, respectively (Supplementary Figs. 40, 41; Supplementary Table 8). This demonstrates that the acyl chain of octanoic acid has undergone a 1,2-rearrangement and is therefore in full agreement with the EMC-like pathway shown in Fig. 6.

## Discussion

Our investigations into the *act* and *mat* BGCs provides key information about the biochemical origins of the rare acyl branched warhead found in both molecules, as well as the hypervariable C-terminus of matlystatins. Following heterologous expression of the *mat* cluster, a gene deletion demonstrated that MatG, a decarboxylase-dehydrogenase similar to epoxyketone synthases, is responsible for the generation of a vinyl ketone that can react with nucleophiles. This represents a new function for this class of proteins. $^{13}$C-labeled precursor feeding revealed that the vinyl ketone derives from methylmalonyl-CoA and that the unusual pyrazole of matlystatins D and F originates from Pip. Mass spectral networking and nucleophile feeding showed that this vinyl ketone is modified by a diverse array of nucleophiles, which hints at a spontaneous non-enzymatic reaction. The ability to exogenously feed synthetic nucleophiles to generate novel matlystatin congeners affords a promising route to new protease inhibitors.

A comparative analysis of the *act* and *mat* gene clusters has provided a model for the biosynthesis of the conserved warhead, and $^{13}$C-labeling was used to support the proposal that it is derived from octanoic acid via an unprecedented variation of the EMC pathway. The proposed use of a mutase to generate the 2-pentyl branched unit represents a potential new function for the mutase family, and is an example of how common primary metabolic pathways can evolve to generate potent chemotypes in secondary metabolism. Interestingly, genome mining indicates that orphan pathways may feature similar mutase-catalyzed reactions. BLAST analysis of the mutases ActD and MatB revealed a small number of highly similar actinomycete orthologs along with many distantly related proteins from thermophilic archaea and bacteria. Every actinomycete ortholog was encoded within a potential secondary metabolite gene cluster and co-located with a putative EMC pathway similar to the actinonin and matlystatin BGC. Our work therefore paves the way for multiple follow-up studies on these fascinating pathways.

## Methods

**Bacterial strains and general experimental procedures.** Chemical, microbiological, and molecular biological agents were purchased from standard commercial sources. *A. atramentaria* DSM 43919 was obtained from the Leibniz Institute DSMZ-German Collection of Microorganisms and Cell Cultures, and *Streptomyces* sp. ATCC 14903 (NCIMB 8845) was obtained from the NCIMB culture collection (Aberdeen, UK) and LGC Standards (Manassas, Virginia, USA). *A. atramentaria* DSM 43919, *Streptomyces* sp. ATCC 14903 (NCIMB 8845), *Streptomyces albus* J1046, *S. coelicolor* M512, *S. coelicolor* M1154, and their respective derivatives were maintained and grown on either MS agar (2% soy flour, 2% mannitol, 2% agar; components purchased from Carl Roth), SMASH agar (instant potato mash agar; 2% Smash powder (Premier Foods), 2% agar), GYM agar (0.4% D-glucose, 0.4% yeast extract, 1% malt extract, pH 7.2), or TSB medium (Becton Dickinson, Heidelberg, Germany). Strains were stored as either frozen mycelium or as spore suspensions in 20 or 30% glycerol at −80 °C. *Escherichia coli* strains were cultivated in LB medium (components purchased from Carl Roth), supplemented with appropriate antibiotics. DNA isolation and manipulations were carried out according to standard methods for *E. coli*[38] and *Streptomyces*[39].

**Production of matlystatins.** About 50 mL TSB broth was inoculated with 100 μL glycerol stock of *A. atramentaria* and incubated for 2 days at 28 °C and 200 rpm. About 1 mL of the culture was transferred to 50 mL of MBG3-7m medium (30 g/L glucose, 70 g/L glycerol, 10 g/L polypepton, 10 g/L soy bean meal, 10 g/L corn steep liquor, 5 g/L MgSO$_4$, and 5 g/L NaNO$_3$)[3] and incubated for 7 days at 28 °C and 200 rpm. The culture broth was centrifuged and the supernatant was extracted with an equal volume of butanol. The organic layer was recovered and the solvent was evaporated under reduced pressure. The remaining residue was dissolved in 2 mL of DMSO and stored at −20 °C.

**Analysis of culture extracts.** Method 1: DMSO solubilised culture extracts were diluted with 9 vol. of methanol and 5 μL thereof were injected onto a Nucleosil-100 C18 reversed-phase HPLC column (3 μm; 100 × 2 mm i.d. fitted with a precolumn 10 × 2 mm, Dr. Maisch GmbH, Ammerbuch, Germany) coupled to a mass spectrometer with an electrospray ionization interface (ESI) (LC/MSD Ultra Trap System XCT 6330; Agilent 1200 series; Agilent Technologies). Chromatography was carried out at a flow rate of 0.4 mL/min with a linear gradient from 10% to 100% solvent B over 30 min (extracts from *A. atramentaria*) or 25 min (*Streptomyces* surrogate hosts) (solvent A: 0.1% (v/v) formic acid (FA) in water; solvent B: 0.06% (v/v) FA in methanol). For MS analysis, ESI (positive and negative ionization) was performed in Ultra Scan mode with a capillary voltage of 3.5 kV and drying gas temperature of 350 °C. The nebulizer pressure was 40.0 psi, drying gas flow was set to 12 L/min. The capillary exit voltage was 121 V, skimmer voltage was 40 V. The trap drive was set to 41.4. Relevant MS data have been deposited in the Mass Spectrometry Interactive Virtual Environment (MassIVE) at UCSD (http://massive.ucsd.edu) and made publicly available (MSV000081442, MSV000081443, and MSV000081445).

Method 2: Samples containing a mixture of mycelia and supernatant were taken from production cultures and 1 volume of methanol was added. Samples were shaken for 30 min, centrifuged at 16,800x*g* for 5 min and the supernatant was used for analysis. LC-MS spectra were obtained using a Shimadzu Nexera X2 UHPLC coupled to a Shimadzu IT-TOF mass spectrometer with ESI. Samples (5 μL) were injected onto a Phenomenex Kinetex 2.6 μm C18 column (50 × 2.1 mm, 100 Å), eluting with a linear gradient of 5–95% acetonitrile in water + 0.1% FA over 6 min with a flow rate of 0.6 mL/min. Positive mode mass spectrometry data were collected between $m/z$ 200 and 1200 with an ion accumulation time of 20 ms featuring an automatic sensitivity control of 70% of the base peak. The curved desolvation line temperature was 300 °C, the heat block temperature was 250 °C, and the nebulizing gas flow was 1.5 L/min. MS$^2$ and MS$^3$ acquisition used a collision-induced dissociation energy of 50% and a precursor ion width of 3 Da. Negative mode MS analysis was carried out using the same parameters as positive mode acquisition. For both positive and negative mode analysis, the instrument was calibrated using sodium trifluoroacetate cluster ions prior to every batch run.

**Mass spectral networking analysis of matlystatin congeners.** The following production media were tested for matlystatin production: (a) MI medium (modified actinonin production medium; 1% glucose, 1% DIFCO soluble starch, 2% corn liquor step, 2% soy flour, 0.25% NH$_4$Cl, 0.3% NaCl and 0.6% CaCO$_3$, adjusted to pH 6.2 with NaOH); (b) MPM medium (matlystatin production medium; 3% D-glucose, 7% glycerol, 1% bacto peptone, 1% soy flour, 1% corn steep liquor, 0.5% MgSO$_4$, 0.5% NH$_4$NO$_3$, 0.5% NaCl); (c) BPM medium (bottromycin production medium; 1% D-glucose, 1.5% starch, 0.5% yeast extract, 1% soy flour, 0.5% NaCl, 0.3% CaCO$_3$, pH 7), and (d) TSB medium. Milli-Q water was used for the preparation of all media. Seed cultures were prepared by inoculating 25 mL TSB medium with frozen mycelia of *A. atramentaria* DSM 43919 in a 250 mL flask containing a coil, and then incubating at 28 °C and 250 rpm for 48 h. About 500 μL of seed cultures was used to inoculate 7.5 mL production medium in 50 mL falcon tubes with the caps replaced with foam bungs. These were incubated at 28 °C with shaking at 250 rpm. Samples containing a mixture of mycelia and supernatant were taken from production cultures and 1 volume of methanol was added. The samples

were shaken for 30 min, spun down at 13,400 rpm for 5 min and then frozen at −20 °C prior to analysis by LC-MS.

LC-MS spectra were obtained using the same LC-MS parameters as defined for "Method 2" section above, where MS$^2$ data were collected in a data-dependent manner for the most abundant species between $m/z$ 200 and 1200, with an exclusion time of 1 s for a given species, where the MS-MS$^2$ cycle period was 0.26 s. LC-MS$^2$ data from duplicate cultures of *A. atramentaria* in MPM, BPM, and TSB were used to construct mass spectral networks using the Global Natural Products Social Molecular Networking server (GNPS, http://gnps.ucsd.edu). Here, the data were clustered with MS-Cluster with a parent mass tolerance of 0.5 Da and a MS$^2$ fragment ion tolerance of 0.5 Da to create consensus spectra. The following settings were used for network construction: minimum pair cosine = 0.65, minimum matched fragment ions = 3, minimum cluster size = 1, minimum peak intensity = 25. The spectra in the resulting networks were then searched against all GNPS spectral libraries available at the time of analysis. Matches were made between network spectra and library spectra, where a score was above 0.7 and there were at least six matched peaks. The spectral files (converted to mzXML format) used in the GNPS analysis are deposited as a MassIVE data set and made publicly available (MSV000081501) and the GNPS analysis is found at http://gnps.ucsd.edu/ProteoSAFe/status.jsp?task = 876a7eb59a9940079944549a810335c3.

Network data were visualized using Cytoscape 2.8.3. Nodes were removed if they were erroneously duplicated by GNPS (same retention time, mass, and MS$^2$ spectrum). The peak areas of extracted ion chromatograms for each node in the matlystatin network were calculated using Shimadzu Quant Browser as the average of the duplicate data and imported into Cytoscape. The square root of these areas was used to define node diameter as a linear gradient from 10 points ($\sqrt{area} = 0$) to 100 points ($\sqrt{area} = 9200$). Following identification by mass spectral networking, putative matlystatin congeners were further characterized by targeted MS$^3$ analysis of MS$^2$ fragments. MS$^3$ data were collected in a data-dependent manner for the most abundant MS$^3$ fragment between $m/z$ 90 and 2000, with an exclusion time of 10 s for a given species, where the MS-MS$^2$- cycle period was 0.47 s. A collision-induced dissociation energy of 50% and a precursor ion width of 3 Da was used. Where necessary, preferred ions for MS$^2$ and/or MS$^3$ fragmentation were prioritized prior to analysis.

## Generation and screening of the fosmid library.
For the construction of the fosmid library high-molecular-weight genomic DNA from *A. atramentaria* was prepared according to standard protocols. The DNA was sheared to ~40-kb fragments and cloned into pCC1FOS (Epicentre Biotechnologies). A library with ~1000 clones was generated in *E. coli* EPI300 according to the manufacturer's instructions. To identify fosmids containing the putative BGC, the library was screened by PCR with primer pair mat_f CTGGTCATGAAGAGACTCGC/mat_r CAGCGACGTGATGTCCTTCG. The primer pair was designed to amplify a 0.6 kb fragment within *matG*. Positive fosmids were characterized by restriction analysis and end-sequencing.

## Bioinformatic analysis of the gene clusters.
antiSMASH (http://antismash.secondarymetabolites.org)[41]. was used to analyze the whole genome sequences of both strains to identify BGCs. Additional analysis and annotation was done using BLAST (http://blast.ncbi.nlm.nih.gov/)[42]. and EMBOSS Needle (http://www.ebi.ac.uk/Tools/psa/emboss_needle/)[43]. The sequence for the *mat* gene cluster can be found at NCBI, Accession Number NZ_KB907224 (G339_RS0128955 - G339_RS37200). The sequence for the *act* gene cluster has been deposited at NCBI under Accession Number KY906183. Detailed information on the gene clusters is available on the Minimum Information about a Biosynthetic Gene cluster (MIBiG) server[44].

## Heterologous expression of the *mat* gene clusters.
For stable chromosomal integration in the *Streptomyces* genome *cat* in fosmid 7C11 was replaced with an integration cassette (int_neo) by λ-Red–mediated recombination to generate fosmid matDK01. The int_neo cassette contains the *int* gene and the attP site of bacteriophage ΦC31 to allow site-specific integration into most *Streptomyces* chromosomes. The cassette was obtained as an *XbaI* restriction fragment from epnLK01[22]. The resulting fosmid matDK01 was verified by restriction analysis and transferred into *E. coli* ET12567[45]. Introduction in *S. coelicolor* M512, *S. albus* J1074, and *S. coelicolor* M1154 was achieved by triparental intergeneric conjugation with the help of *E. coli* ET12567/pUB307[46]. Exconjugants resistant to kanamycin were selected and designated as *S. coelicolor* M512/matDK01 (1–3), *S. coelicolor* M1154/matDK01 (1–3), and *S. albus* J1074/matDK01 (1–3). For the production of matlystatins in *Streptomyces*, 50 mL TSB broth were inoculated with 10 μL spore suspension and incubated for 2 days at 30 °C and 200 rpm. About 1 mL of the preculture was transferred to 50 mL of R5 medium (103 g/L sucrose, 0.25 g/L K$_2$SO$_4$, 10.12 g/L MgCl$_2$·6H$_2$O, 10 g/L glucose, 0.1 g/L Casaminoacids, 5 g/L yeast extract, 5.73 g/L TES (*N*-tris(hydroxymethyl)methyl-2-aminoethanesulfonic acid), 80 μg/L ZnCl$_2$, 400 μg/L FeCl$_3$·6H$_2$O, 20 μg/L CuCl$_2$·2H$_2$O, 20 μg/L MnCl$_2$·4H$_2$O, 20 μg/L Na$_2$B$_4$O$_7$·10H$_2$O, 20 μg/L (NH$_4$)$_6$Mo$_7$O$_{24}$·4H$_2$O, 50 mg/L KH$_2$PO$_4$, 3 g/L L-proline, 2.94 g/L CaCl$_2$, and 280 μg/L NaOH) and incubated for 7 days at 30 °C and 200 rpm. Extraction procedure of cultures was performed as described in the

"Production of matlystatins" section and LC-MS analysis was carried out as described in the "Analysis of culture extracts" section.

## Generation of gene deletion mutants.
In-frame gene deletion mutants were carried out in *S. coelicolor* M1154. An apramycin resistance cassette [*aac(3)IV*] was amplified from plasmid pIJ773 by PCR with primer pairs orf-3_F and orf-3_R for the inactivation of *orf-3*, orf-2_F, and orf-2_R for the inactivation of *orf-2*, orf-1_F, and orf-1_R for the inactivation of *orf-1*, matA_F, and matA_R for the inactivation of *matA*, matB_F, and matB_R for the inactivation of *matB*, matC_F, and matC_R for the inactivation of *matC*, matE_F, and matE_R for the inactivation of *matE*, matG_F, and matG_R for the inactivation of *matG*, matH_F, and matH_R for the inactivation of *matH*, matJ_F, and matJ_R for the inactivation of *matJ*, matK_F, and matK_R for the inactivation of *matK*, matL_F, and matL_R for the inactivation of *matL*, matQ_F, and matQ_R for the inactivation of *matQ*, matR_F, and matR_R for the inactivation of *matR*, orf+1_F, and orf+1_R for the inactivation of *orf+1*, orf+2_F, and orf+2_R for the inactivation of *orf+2* and orf+3_F, and orf+3_R for the inactivation of *orf+3* (for primer list, see Supplementary Table 4). The ORFs and genes, respectively, were replaced in *E. coli* BW25113/pKD46/matDK01 applying the PCR targeting system[23]. The resulting mutant fosmids were confirmed by restriction analysis. To generate precisely tailored in-frame mutations, the resistance cassette was removed in *E. coli* BT340, taking advantage of the flanking Flp/FRT recognition site. Positive fosmids were screened for their apramycin sensitivity and verified by restriction analysis, PCR, and sequencing of PCR products (for primer list, see Supplementary Table 4). Fosmids matFL01 (Δ*orf-3*), matFL02 (Δ*orf-2*), matFL03 (Δ*orf-1*), matJZ01 (Δ*matA*), matFL04 (Δ*matB*), matPB01 (Δ*matC*), matJZ02 (Δ*matE*), matFL06 (Δ*matG*), matDS01 (Δ*matH*), matDS02 (Δ*matJ*), matDS03 (Δ*matK*), matPB02 (Δ*matL*), matPB03 (Δ*matQ*) matFL07 (Δ*matR*), matFL08 (Δ*orf+1*), matFL09 (Δ*orf+2*), and matFL10 (Δ*orf+3*) were transferred into *E. coli* ET12567[9] and introduced into *S. coelicolor* M1154 by triparental intergeneric conjugation with the help of *E. coli* ET12567/pUB307[46]. For every mutation three individual kanamycin resistant clones were selected and designated as *S. coelicolor* M1154/matFL01 (1–3; Δ*orf-3*), *S. coelicolor* M1154/matFL02 (1–3; Δ*orf-2*), *S. coelicolor* M1154/matFL03 (1–3; Δ*orf-1*), *S. coelicolor* M1154/matJZ01 (1–3; Δ*matA*), *S. coelicolor* M1154/matFL04 (1–3; Δ*matB*), *S. coelicolor* M1154/matPB01 (1–3; Δ*matC*), *S. coelicolor* M1154/matJZ02 (1–3; Δ*matE*), *S. coelicolor* M1154/matFL06 (1–3; Δ*matG*), *S. coelicolor* M1154/matDS01 (1–3; Δ*matH*), *S. coelicolor* M1154/matDS02 (1–3; Δ*matJ*), *S. coelicolor* M1154/matDS03 (1–3; Δ*matK*), *S. coelicolor* M1154/matPB02 (1–3; Δ*matL*), *S. coelicolor* M1154/matPB03 (1–3; Δ*matQ*), *S. coelicolor* M1154/matFL07 (1–3; Δ*matR*), *S. coelicolor* M1154/matFL08 (1–3; Δ*orf+1*), *S. coelicolor* M1154/matFL09 (1–3; Δ*orf+2*), and *S. coelicolor* M1154/matFL10 (1–3; Δ*orf+3*). Cultivation, extraction, and analysis were conducted as described for *Streptomyces* strains containing the intact cluster.

## $^{13}$C-labeled precursor feeding in *A. atramentaria*.
For feeding experiments with [2-$^{13}$C]propionate and [U-$^{13}$C$_5$]-L-ornithine (both purchased from Sigma Aldrich) in *A. atramentaria*, matlystatins were produced as described in "Production of matlystatins" section.: After 1 day of cultivation in MBG3-7m media, isotope-labeled precursors were aseptically added to each culture in final concentrations of 5 mM (or 4 mM for preparative scale) [2-$^{13}$C]propionate and 1.5 mM [U-$^{13}$C$_5$]-L-ornithine, respectively. LC-MS analysis was carried out as described in "Analysis of culture extracts" section.

## Purification of matlystatins.
Deshydroxymatlystatin A was purified from 1 L culture containing 4 mM [2-$^{13}$C] propionate and deshydroxymatlystatins A, B, and D/F from 1 L without supplementation. Cultivation was performed as described above. The culture broth was centrifuged and the supernatant was extracted with a 1.5-fold volume of butanol. The organic layer was recovered and the solvent was evaporated under reduced pressure. The remaining residue was dissolved in 6 mL DMSO and diluted with 30 mL methanol. The supernatant was separated from the precipitate by centrifugation and the methanol was evaporated under reduced pressure. The remaining extract was diluted with 30 mL acetonitrile and insoluble substances were eliminated. The extract was concentrated and applied to a Reprosil C18 reversed-phase HPLC column (250 × 10 mm, 5 μm; fitted with a precolumn 30 × 10 mm; Dr. Maisch GmbH, Ammerbuch, Germany). Chromatography was carried out at an Agilent 1200 series system (Agilent Technologies) at a flow rate of 4 mL/min with a linear gradient from 10 to 45% solvent B over 10 min, 45 to 55% solvent B over 16 min, and 55 to 100% solvent B over 5 min (solvent A: 0.1% (v/v) FA in water; solvent B: 0.06% (v/v) FA in acetonitrile). UV detection was performed at 210 nm. $^{13}$C-Deshydroxymatlystatin A eluted at 15.5 min and was evaporated to dryness in vacuo to yield 18.3 mg/L. Deshydroxymatlystatins were recovered in yields of 33 mg/mL (A), 3 mg/mL (B), and 8 mg/mL (D/F).

## NMR analysis of $^{13}$C-deshydroxymatlystatin A.
1D and 2D NMR spectra were recorded at a $^1$H resonance frequency of 400 MHz using a Bruker Avance III HD 400 NMR spectrometer, equipped with a 5 mm broadband fluorine observer (BBFO) probe head. Spectra were calibrated to the residual solvent signals of $d_7$-N, N-dimethyl-formamide with resonances at $\delta_{H/C}$ = 2.92/34.89. For the calculation of the enrichment factor, see Supplementary Table 5.

**Production and purification of His-matlystatin derivative 6**. About $10 \times 2$ L flasks, each containing 200 mL of MPM, were each inoculated with 10 mL of an *A. atramentaria* DSM 43919 seed culture grown for 72 h in TSB medium, as described before. These production cultures were incubated at 28 °C with shaking at 250 rpm. Starting at 72 h, each culture was fed every 24 h with 20 mL of a histidine solution (250 mM in water, sterilized by filtration). After 216 h fermentation, culture broths (~3 L after His feeding) were extracted with methanol (3 L). The mycelium was separated by filtration to yield a cell-free supernatant (ca. 6 L). The mycelial cake was further extracted with methanol (3 L) and harvested by centrifugation. The supernatant and the mycelial cake extracts were combined (9 L) and then concentrated to 3 L under reduced pressure. The resulting aqueous solution was extracted with ethyl acetate ($3 \times 3$ L) and then with 1-butanol ($3 \times 3$ L) and then the solvent was removed from each extract to afford an ethyl acetate extract (A, 3.5 g) and a butanol extract (B, 23.0 g). LC-MS analysis determined that the target compounds were mainly in the butanol extract. About 16 g of the butanol extract was subjected to vacuum liquid chromatography (VLC) on C18 RP using a gradient of $H_2O$:MeOH (100:0 to 0:100). Fractions containing the targeted matlystatin derivatives were further fractionated on a Sephadex LH-20 column using MeOH/$H_2O$ (7:3) as the mobile phase. The major matlystatin-containing fraction was subjected to reverse phase preparative HPLC (Gemini NX-C18, 150 mm × 21.2 mm, 5 μ (7:3)) as the mobile phase. The major matlystatin-containing fraction/$H_2O$ (+0.1% trifluoroacetic acid) ranged from 10 to 45% MeCN over 45 min. The fraction containing **6** was further purified by using a semipreparative HPLC column (Phenomenex, Luna PFP(2), 250 mm × 10 mm, 5 μ was further purified by using a semipreparative HPLC column × 1 MeCN/$H_2O$ (+0.1% FA) from 10 to 35% MeCN over 45 min yielding **6** (0.4 mg, $t_R$ 35.5 min). 1D and 2D NMR spectra (Supplementary Figs. 7–12) were recorded at a $^1$H resonance frequency of 600 MHz and a $^{13}$C resonance frequency of 150 MHz using a Bruker Avance 600 MHz NMR spectrometer operated using Topspin 2.0 software. Spectra were calibrated to the residual solvent signals of $CD_3OD$ with resonances at $\delta_H$ 3.31 and $\delta_C$ 49.0.

**Nucleophile feeding to *A. atramentaria***. Stock solutions of tranexamic acid (TXA, 500 mM in water), thioglycolic acid (TGA, 1 M in water), methyl thioglycolate (MTG, 1 M in ethanol), and 4-bromobenzyl mercaptan (BBM, 1 M in ethanol), all purchased from Sigma Aldrich, were prepared and sterilized by filtration. Cultures of *A. atramentaria* in MPM were fed every 24 h, starting at 72 h and finishing at 192 h. A variety of nucleophile amounts were assayed for each feeding experiment, with the daily addition of a final concentration of 1, 2, 5, 10, 20, and 50 mM tested for TXA, TGA, and MTG, and 1, 10, and 50 mM for BBM. Extraction of metabolites and LC-MS analysis was carried out as described for LC-MS "Method 2" section.

**Genome sequencing of *Streptomyces* strain ATCC14903**. Isolation of genomic DNA of *Streptomyces* sp. ATCC14903 (NCIMB 8845) was following standard operating protocols[2]. The quality of the purified DNA was assessed by gel electrophoresis and the quantity was estimated by fluorescence-based methods using Quant-iT PicoGreen dsDNA kit (Invitrogen) and Tecan Infinite 200 Microplate Reader (Tecan Deutschland GmbH) analysis. An 8K mate pair library (Nextera Mate Pair Sample Preparation Kit, Illumina) and a whole-genome shotgun PCRfree library (Nextera DNA Sample Prep Kit, Illumina) were prepared according to the manufacturer's instructions. The *Streptomyces* sp. ATCC14903 (NCIMB 8845) library was sequenced using an Illumina MiSeq system (2 × 300 bases). For de novo assembly, a GS De Novo Assembler software release version 2.8 (Roche) was used for sequencing and processing the raw data. *Streptomyces* sp. ATCC14903 (NCIMB 8845) has a draft genome sequence with a GC content of 73.35% containing 6882 protein coding regions and a total size of 8,457,327 bp. A plasmid with a size of 13,210 bp and a GC content of 70.03% was also sequenced. Details will be reported elsewhere.

**Isotopic labeling with [1,2,3,4-$^{13}$C$_4$]octanoic acid**. Seed cultures were prepared by inoculating 25 mL TSB medium with frozen mycelia of *A. atramentaria* DSM 43919 and *Streptomyces* sp. NCIBM 8845 in a 250 mL flask containing a coil, and then incubating at 28 °C and 250 rpm for 48 h. About 0.5 mL of the *Streptomyces* NCIBM 8845 culture was used to inoculate a 25 mL flask containing 5 mL of MI medium, and 0.5 mL of the *A. atramentaria* DSM 43919 seed culture was used to inoculate a 25 mL flask containing 5 mL of MPM. These were incubated at 28 °C with shaking at 250 rpm.

Stable isotope labeling experiments were performed by feeding a 1 mM final concentration of [1,2,3,4-$^{13}$C$_4$]octanoic acid (Cambridge Isotope Laboratories, USA) every 24 h to the production cultures of both *Streptomyces* NCIBM 8845 and *A. atramentaria*, starting at 48 h and finishing at 196 h. Unlabeled octanoic acid (Sigma Aldrich) was added in the same way to control cultures. Both unlabeled and labeled octanoic acid were pre-dissolved in ethanol (15% v/v) to facilitate its homogeneous diffusion in the culture. In the case of *Streptomyces* sp. NCIBM 8845, six 5 mm glass beads (Sigma Aldrich) were added to the culture flask from the beginning of the fermentation to facilitate disperse growth. This was to avoid the formation of large mycelial clumps that would hamper the diffusion of the fed octanoic acid across the cell wall. LC-MS spectra were obtained as described for

mass spectral networking analysis, with additional negative mode MS$^2$ data collected for actinonin using the same LC conditions.

**Purification and NMR analysis of $^{13}$C$_4$-actinonin**. To obtain sufficient $^{13}$C$_4$-actinonin for NMR characterization, a total of $12 \times 100$ mL flasks each containing 20 mL of MI and $12 \times 5$ mm glass beads were inoculated with 2 mL of a *Streptomyces* sp. NCIBM 8845 seed culture. The seed culture preparation, production culture conditions, and $^{13}$C$_4$-octanoic acid feeding were performed as described above. The combined culture broths were extracted with 0.2 L of methanol and the methanolic mixture was harvested by centrifugation. Removal of the solvent under reduced pressure yielded 4.84 g of methanolic extract, which was fractionated by VLC on C18-reverse phase resin using mixtures of $H_2O$/MeOH (0–100% MeOH) as eluent. Following removal of methanol, the actinonin-containing fraction was subjected to a liquid–liquid extraction with butanol (x3) and the butanolic fraction was fractionated on a Sephadex LH-20 column using a mixture of MeOH/$H_2O$ (3:1) as the mobile phase. This was further purified on a semipreparative HPLC column (Phenomenex Luna PFP (2), 250 mm × 10 mm, 5 μm; 3 mL/min, UV detection at 200 nm) using $CH_3CN$/$H_2O$ as a mobile phase, with a linear gradient of 5–40% $CH_3CN$ over 40 min yielding pure labeled actinonin (0.1 mg, $t_R$ 35.8 min). 1D and 2D NMR spectra were recorded at a $^1$H resonance frequency of 600 MHz and $^{13}$C resonance frequency of 150 MHz using a Bruker Avance 600 MHz NMR spectrometer operated using Topspin 2.0 software. Spectra were calibrated to the residual solvent signals of $CD_3OD$ with resonances at $\delta_H$ 3.31 and $\delta_C$ 49.0.

**Data availability**. The authors declare that the data supporting the findings reported in this study are available within the article and Supplementary Information, or are available from the authors on reasonable request. New nucleotide sequence data have been deposited in NCBI GenBank under the accession number KY906183 (*act* BCG). Both *act* and *mat* BGCs have been submitted to the MIBiG (Minimum Information about a Biosynthetic Gene cluster) repository: BGC0001442 (*act*) and BGC0001443 (*mat*). Mass spectral data have been deposited at the online platform MassIVE (Mass Spectrometry Interactive Virtual Environment; MSV000081442, MSV000081443, MSV000081445, MSV000081501). The GNPS analysis can be found at http://gnps.ucsd.edu/ProteoSAFe/status.jsp?task=876a7eb59a9940079944549a810335c3

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

## Acknowledgements

This work was supported by the German Research Foundation (DFG KA 3071/4-1 to L. K.), a Royal Society University Research Fellowship (A.W.T.), an LGFG grant at the University of Tübingen (F.W.), the SFB 766 (A.K.), and the BBSRC MET ISP (BB/J004561/1) at the John Innes Centre (A.W.T. and J.S.-A.).

## Author contributions

A.W.T. and L.K. supervised the study. F.L., J.S.-A., A.W.T., and L.K. designed the study with inputs from all authors. F.L., D.P.S., D.K., and P.B. performed genetic experiments. F.L., J.S.-A., D.P., and D.P.S. cultivated strains for analytical studies and compound isolation, and performed feeding experiments. F.W. prepared sequencing of *Streptomyces* ATCC 14903 (NCIMB 8845) genome. A.K., A.W.T., J.S.-A., and D.P. performed LC-MS analysis. N.K. and R.L. performed NMR spectroscopy. F.L., H.G., A.W.T., and L.K. wrote the manuscript. All authors commented on and edited the manuscript.

## Additional information

**Competing interests:** The authors declare no competing financial interests.

