## [Peer Review File · Nature Communications]

Reviewers' comments:

Reviewer #1 (Remarks to the Author):

This manuscript describes the identification of the biosynthetic gene clusters for two important natural products, actinonin and matlystatin, both having promising bioactivity as metalloproteinase inhibitors and sharing a hydroxamate moiety as the warhead. Carefully designed studies including bioinformatics analysis, precursor feeding studies, heterologous expression and selected gene inactivation provided evidence to validate the proposed gene cluster and biosynthetic pathway for matylstatins. Interesting biosynthetic features of matylstatins include a modified ethylmalonyl-CoA pathway to generate the N-hydro-alkyl-succinamic acid warhead, and the flexibility of a flavin-dependent decarboxylase-dehydrogenase homolog in generating structural diversity at the C-terminus of matlystatins. However, the conclusions here have not been fully supported by data, and the presentation could also be improved for clarity. Here are a few specific comments:

1. Although metabolomics analysis coupled with precursor feeding experiments is a valid method to probe the biosynthetic pathway, the assignment of each enzyme activity is too preliminary, in particular in a high-impact journal. Since the gene knockout will not be informative in this case, when possible, enzyme activity reconstitution in vitro should be included, such as the pathway shown in Fig6a.
2. The titer of all compounds should be reported.
3. Which enzyme, MatD or MatA, is proposed to be responsible for hydroxamate formation? MatA is indicated in Fig6A and line 110, while MatD is indicated in line 142 and 331 and also to be involved in piperazic acid formation.
4. What is the reason for only 1a and 2a production in heterologous hosts? Does this mean the gene cluster is insufficient for 3-5 biosynthesis?
5. In heterologous expression, was the titer of 2a significantly increased in matG mutant compared to wildtype? I would guess so because MatG is competing with the spontaneous formation of 2a (line 152 and Fig. 3B).
6. The NMR of 1a (table S4 and line 193) indicates that there are two conformers in a ratio of 3:2. Please explain the difference of the conformers.
7. The structures of novel deshydroxymatlystatins need more evidence. The LC/MS/MS is not sufficient to draw the structures, and the authors need to provide NMR for at least one of the novel compounds.
8. Please also include structures of deshydroxy amide derivatives of matlystatins in Fig 1. In Fig4, please avoid text/figure overlaps. Please also include the bioinformatics analysis of the

substrate specificity of AT domain of MatO to support the methylmalonyl-CoA specificity.

Reviewer #2 (Remarks to the Author):

The major claims of the paper are that 1. matlystatin biosynthesis requires an EpnF homologue with novel "partial" activity that generate an alpha-beta unsaturated ketone that can serve as a Michael acceptor to make new molecules; 2. matlystatin hydroxamate warhead is assembled through a mutase-required pathway that makes alkyl substituted succinyl-CoA derivatives for incorporation into matlystatin and 3. actinonin uses a similar mutase-required pathway to make similar alkyl-substituted succinyl-CoA precursors. The manuscript essential includes two genome mining results: the matlystatin biosynthetic gene cluster (BGC) was identified based on the EpnF homologue, and the actinonin gene cluster was identified based on characterization of the matlystatin BGC. The data are solid, and convincing evidence is provided to support the conclusions. Overall the results are very interesting and significant as it defines a new way to make natural products and sets the stage for complete pathway delineation included the exciting enzymology associated with making the succinyl-incorporated units. The only comments or suggested changes have to do with the organization or terminology as follows:

1. The manuscript is somewhat difficult to follow, particularly the introduction. The first couple of paragraphs of the Introduction emphasize the natural product actinonin, yet most of the manuscript is experimental data for the matlystatin natural products. It is also not obvious, based on structural inspection, why actinonin would have a homologue of the ACAD EpnF from the eponemycin gene cluster. One possibility for making the introduction easier to follow is to start with line 62 and integrate the first two paragraphs into the results and discussion, where appropriate. This would allow for the authors to flesh-out more of the details of gene-oriented approach to linking biosynthetic gene clusters to natural products with unique chemistry. Line 70 "We used EpnF" could start a new paragraph to detail how the intro of the EpnF enzyme was used to guide this study.
2. In the Introduction, Line 74, a putative ethylmalonyl-CoA pathway is eluded to. What is the ethylmalonyl-CoA pathway and how does it relate to matylstatins? I do not see any ethylmalonyl-CoA as a precursor in any of these molecules. Is this the pathway that involves a CCR and mutase to make the alkyl-substituted succinyl-CoA precursor? Perhaps this could be better described in the introduction for clarity.
3. Line 116, the compounds that are produced are described as the deshydroxy amide derivatives, which is confusing. The compounds could be better described as either deshydroxy-matlystatins or the terminal amide derivatives.
4. Line 134, not really sure what is meant by mutant strains. Is mutant strain referring to the strain with the inserted fosmid containing the unknown biosynthetic gene cluster?

Reviewer #3 (Remarks to the Author):

About the study findings:

In this manuscript, Leipoldt and Santos-Aberturas et al characterized the biosynthetic gene clusters (BCG) of the pathway for matlystatins in *Actinomadura atramentaria*, along with the biosynthetic line of actinonin, another metalloprotease inhibitor. The authors first found an orphan pathway in *A. atramentaria* using AntiSmash/BLAST encoding for a hybrid NRPS/PKS assembly line. Mass spectrometry experiments and previous studies on this genus guide allowed the authors to annotate the presence of known matlystatins in the culture. These results were consolidated using detailed tandem mass-spectrometry, feeding experiments of labeled precursors, and ultimately the isolation of one derivative and its structural characterization by NMR. In addition, unknown matlystatin analogues were detected using mass spectrometry molecular networking.

The authors then studied in detail the biosynthetic assembly lines of metalloproteinase inhibitors using knock-out mutants and mass spectrometry associated with feeding experiments. Finally, novel matlystatin congeners were generated using exogenously fed synthetic nucleophiles.

Main comments

I recommend the manuscript for publication in *Nature Communication* without major concerns. First of all, this study describes for the first time the biosynthetic pathway of actinonin and matlystatin, members of hydroxamate metalloproteinase inhibitors, which paves the way to the potential discovery of future drugs by (1) genome-mining in other species, and (2) production of a diversity of analogues by bio-engineering.

Interestingly, the authors employed a remarkable panel of complementary approaches/tools, including advanced experiments (feeding, knockout) and manual annotation (MS/MS fragmentation) along with bioinformatics resources (AntiSmash/Blast, GNPS), which allowed them to characterize the biosynthetic assembly lines in metalloproteinase inhibitors. It really represents a state of art example for the elucidation of uncharacterized BCG.

The results are discussed consciously in the manuscript and interpretation/hypothesis are supported by the data (exposed in detail in the SI). And the manuscript is very well written and argumentation is flowing nicely.

My main concern is related to MS data reproducibility and open science consideration.

+Data reproducibility:

++In SI, material and methods: provide detailed informations about source parameters for the two mass spectrometers, and their data dependent methods properties.

++Mass spectrometry data: LC-MS(-MS) data must be uploaded to a public repository such as Metabolights (<http://www.ebi.ac.uk/metabolights/>) or MassIVE (<http://massive.ucsd.edu>).

Indicate that in the manuscript, and provide the deposit number.

++Molecular networking: Important provide in the manuscript or SI the url link to the GNPS molecular networking job. Provide the Cytoscape file as supporting information.

+ Open science:

- Mass spectrometry: To promote data sharing with the community, authors must deposit the MS/MS spectra of matlystatins and congeners in public spectral library. This can be achieved easily using the GNPS molecular networking job (clic on View Cluster Spectra, and annotate to spectral library). In addition, this will allow the researchers to know automatically if there is any other public MassIVE dataset that would contain these compounds or related ones. Indicate that in the manuscript, and provide the deposit numbers.

- Materiel and methods about MS: please provide more detailed informations about source parameters, and data dependent methods properties.

Minor comments:

+Abstract.

-Articulation between sentences 1 and 2 is not obvious. It would beneficial to present that actinonin is a mettalo-proteinase inhibitor.

Introduction:

- Line 42. Insert a reference for natural products as drug. Such as <http://pubs.acs.org/doi/abs/10.1021/acs.jnatprod.5b01055>

- Line 117: explicit MS2

- Line 220: Instead of "indicates", I would recommend "suggests" as 4 peaks are actually observable in the figure S4.

- Use MS/MS or MS2 throughout the text.

Suggestion:

I would encourage the authors to take full advantage of latest tools in the field, such as the Dereplicator <https://www.ncbi.nlm.nih.gov/pmc/articles/PMC5409158/> which is available on GNPS (in silico workflow), and that would have certainly detected the presence known matlystatins. This would have been done in an automated fashion and would have save a lot of

effort. Because data are not deposited on public repository I could not try myself with the paper data, but I expected that other peptidic-natural products could be detected with that tools. If possible, I would suggest to run a dereplicator job and annotate detected compounds in GNPS spectral library. Note that there is an analogue search mode in the dereplicator.

Response to Reviewers' Comments:

Reviewer 1:

1. *Although metabolomics analysis coupled with precursor feeding experiments is a valid method to probe the biosynthetic pathway, the assignment of each enzyme activity is too preliminary, in particular in a high-impact journal. Since the gene knockout will not be informative in this case, when possible, enzyme activity reconstitution in vitro should be included, such as the pathway shown in Fig6a.*

We agree with the reviewer that the *in vitro* characterization of all biosynthetic enzymes encoded in the two gene clusters would be very interesting. However, this a very comprehensive endeavor that could easily span multiple follow-up papers. As the reviewer mentions, metabolomics-coupled precursor feeding is a valid approach to probe a biosynthetic pathway. In addition, our biosynthetic model is strongly supported by the comparative analysis of the matylstatin and actinonin gene clusters and the amount of existing functional data on homologous enzymes in the literature. We would thus argue that, in particular for the enzymes proposed to be involved in the ethylmalonyl-CoA-like pathway shown in Figure 6A, we have sufficient evidence to allow the assignment of specific functions in the biosynthesis.

Nevertheless, we have carried out *in vitro* work on the putative *N*-hydroxylase ActI. A recent paper (Kudo et. al., JACS 2017) reported that the *N*-hydroxy group in trichostatins is generated via hydroxylation of L-glutamine and subsequent hydroxylamine transfer. We were curious to determine whether *N*-hydroxylation in actinonin and matylstatin biosynthesis is mechanistically equivalent. However, our data suggest that this is not the case, as no hydroxylated glutamine was detected in an *in vitro* assay using ActI, even though a synthetic standard of HO-Gln was detected readily.

Instead, to meet the request of the reviewer and provide further support on our proposed pathway, we isolated $^{13}\text{C}_4$ -actinonin from a fed culture for ^{13}C NMR analysis. The cost of the labelled material prevented the large-scale production of this compound, but the purified material provided selective enrichment of four peaks with shifts and splitting that is entirely consistent with our proposed biosynthetic model. This further supports the involvement of an ethylmalonyl-CoA-like pathway in the formation of actinonin and matylstatin, in particular the unusual mutase-catalyzed step that provides the branched acyl group. Furthermore, we have performed an additional gene knock-out

(*Δorf-1*) to unambiguously determine the cluster boundaries and the repertoire of biosynthetic enzymes. These additional experiments have now been included in the revised manuscript.

2. The titer of all compounds should be reported

As suggested by the reviewer we have included additional information on the yields of actinonin and matlystatin derivatives per litre of culture broth in the revised manuscript. In our feeding experiments to label both actinonin and deshydroxymatlystatin A, we now report yields of 0.42 mg/L and 18.3 mg/L, respectively. In addition, we show that feeding of histidine leads to the isolation of 0.13 mg/L of the His-matlystatin derivative **6**. To further address the concerns of the reviewer we also isolated deshydroxymatlystatins B and D/F from *A. atramentaria* DSM 43919. The following sentence has been included in the Results section of the revised manuscript:

“In order to evaluate the production rates of the deshydroxy matlystatin derivatives we isolated 1a, 2a and 3a/5a from cultures of *A. atramentaria* DSM 43919 yielding 33 mg/L, 3 mg/L and 8 mg/L, respectively.”

Some of the compounds that are produced could not be accurately quantified, given the low levels of molecules that were identified by LC-MS² analysis. For example, the compounds produced by feeding unnatural nucleophiles and some of the derivatives identified by mass spectral networking. Therefore, our use of differentially sized network nodes provides an approximation of the relative quantities of each compound based on signal intensities.

3. Which enzyme, *MatD* or *MatA*, is proposed to be responsible for hydroxamate formation?

We thank the reviewer for pointing that out and want to apologize for the error. *MatA* is proposed to be responsible for hydroxamate formation. We have changed this in our revised manuscript.

4. What is the reason for only 1a and 2a production in heterologous hosts? Does this mean the gene cluster is insufficient for 3-5 biosynthesis?

Our feeding experiments indicate that the relative abundance of the various matlystatins is dependent on the amount of the respective nucleophiles. It is therefore possible that less Pip is produced in the heterologous host, which would lead to lower amounts of *Mat D/F*.

Motivated by the comments of the reviewer we have carefully reanalyzed the heterologous mutants and found the production of deshydroxy derivatives of *Mat D/F* in small amounts only in extracts of the *S. coelicolor* M1154 strains. We have included the LC-MS² analysis to confirm the identity of the compounds as Figure S5 in the Supporting Information.

In the main text the respective section has been modified accordingly and now states:

“The accumulation of deshydroxy derivatives of matlystatin A (1a) and B (2a) was observed for all heterologous mutants (Figure 3B). In addition, we found low amounts of deshydroxymatlystatins D/F only in extracts of the *S. coelicolor* M1154 strains with the cluster (Figure S5). For the following studies we thus used *S. coelicolor* M1154 as a host.”

5. In heterologous expression, was the titer of 2a significantly increased in *matG* mutant compared to wildtype? I would guess so because *MatG* is competing with the spontaneous formation of 2a.

We thank the reviewer for the suggestion. We have checked the metabolic profile of the *matG* mutant and compared it to the wild type strain, and we were not able to find a significant change in the production levels of 2a. In general, we would be very hesitant to draw conclusions from a change in production levels observed in any mutant strains. Often gene knock-outs in actinomycetes, even markerless in-frame deletions, result in unforeseen minor changes to growth or

transcription levels. Comparative quantitative analysis of production rates usually requires standardized cultivation systems and would be beyond the scope of the paper.

6. *The NMR of 1a (table S4 and line 193) indicates that there are two conformers in a ratio of 3:2. Please explain the difference of the conformers.*

We agree with the reviewer that the observation of the conformers is unusual for this class of compounds. The majority of significant ^{13}C NMR chemical shift differences between the two conformers of 1a were observed in the Pip portion. This means that the conformers can be attributed to *cis* and *trans* rotamers of the amide bond between the Pip moiety and the adjacent isoleucine moiety. Commonly, *N*-acyl derivatives of Pip display an elevated degree of conformational rigidity. However, rotamers can be occasionally observed depending on the substituents and applied NMR solvent. Such rotamers have been described e.g. by the Walsh group for synthetically prepared *N*-protected Pip-allyl or methyl esters (Jiang et al, Biochemistry 2011). In the revised manuscript we have added the following sentence to the main text: "... indicating the presence of two conformers in a ratio of 3:2. This observation may be explained by the presence of *cis/trans* rotamers of the amide bond connecting the Pip and the Ile moiety."

7. *The structures of novel deshydroxymatlystatins need more evidence. The LC/MS/MS is not sufficient to draw the structures, and the authors need to provide NMR for at least one of the novel compounds.*

We agree that NMR is required to unambiguously determine the structures of the newly identified matlystatins. Therefore, to address the concerns of the reviewer we conducted a large-scale fermentation of *A. atramentaria* DSM 43919 in matlystatin production medium fed with histidine. Without feeding, the proposed histidine congener could not be detected in this medium, but multiple derivatives of His-matlystatin could be elicited by histidine feeding (Figure 5D). Derivative 6 proved amenable to purification and was structurally characterized by extensive NMR experiments. The respective NMR data has been added to the Supporting Information of the revised manuscript as Table S7 and Figures S24-S30. The main text of the revised manuscript has been modified to include the following text:

"To test this hypothesis, *A. atramentaria* was fermented in MPM supplemented with histidine, which yielded a significant amount of both the putative matlystatin-His and deshydroxymatlystatin-His congeners (m/z 594.36 and 578.36, respectively, Figure 5D and Figure S23), as well as another related compound with m/z 561.34 (**6**). Crucially, without supplementation of histidine, these compounds were not detected in MPM culture extracts (Figure 5A and Figure S23). **6** proved most amenable to purification from a large-scale fermentation and was structurally characterized by MS² and NMR (^1H , ^{13}C , COSY, HSQC, HMBC, NOESY; Figure 5D and Figures S24-S30). This revealed a matlystatin-like compound with a histidine residue attached at the C-terminus and loss of hydroxylamine to yield a fused bicyclic ring. Crucially, HMBC and NOESY correlations between C1'' (4.25 ppm) and C5''' and C6''' on the histidine moiety determined that N $_{\tau}$ of histidine reacts with the vinyl ketone."

8. *Please also include structures of deshydroxy amide derivatives of matlystatins in Fig 1. In Fig4, please avoid text/figure overlaps. Please also include the bioinformatics analysis of the substrate specificity of AT domain of MatO to support the methylmalonyl-CoA specificity.*

We thank the reviewer for the helpful comments. In the revised manuscript Figure 1 and Figure 4 have been modified accordingly. The predicted A- and AT-domain specificities are included in the SI (Table S3). It is worth noting that our feeding experiment with ^{13}C -labeled propionate provide strong evidence for the methylmalonyl-CoA specificity of the MatO AT-domain.

Reviewer 2:

“Overall the results are very interesting and significant as it defines a new way to make natural products and sets the stage for complete pathway delineation included the exciting enzymology associated with making the succinyl-incorporated units.”

We thank the reviewer for the encouraging comments!

1. The manuscript is somewhat difficult to follow, particularly the introduction. The first couple of paragraphs of the Introduction emphasize the natural product actinonin, yet most of the manuscript is experimental data for the matlystatin natural products. It is also not obvious, based on structural inspection, why actinonin would have a homologue of the ACAD EpnF from the eponemycin gene cluster. One possibility for making the introduction easier to follow is to start with line 62 and integrate the first two paragraphs into the results and discussion, where appropriate. This would allow for the authors to flesh-out more of the details of gene-oriented approach to linking biosynthetic gene clusters to natural products with unique chemistry. Line 70 “We used EpnF” could start a new paragraph to detail how the intro of the EpnF enzyme was used to guide this study.

We thank the reviewer for the suggestion. In the revised manuscript we have modified the Introduction section accordingly. The detailed information on hydroxamate protease inhibitors with a focus on actinonin that was provided in the submitted manuscript has now been removed from the Introduction section. Parts of this information has been added to the Results section in a paragraph that follows the discovery of the matlystatin gene cluster.

2. What is the ethylmalonyl-CoA pathway and how does it relate to matlystatins? Perhaps this could be better described in the introduction for clarity.

We would like to apologize for any confusion that we may have caused. In fact, a full ethylmalonyl-CoA (EMC) pathway is not involved in actinonin or matlystatin biosynthesis. However, we find genes in both clusters that show high homology to genes known to take part in the EMC pathway. We have thus stated that an EMC-like pathway might be involved in the assembly of the common *N*-hydroxy-2-pentyl-succinamic acid moiety. In the original manuscript, this was described in detail towards the end of the manuscript on page 11 as part of the gene deletion results section. For clarity, this detail has been moved to the beginning of the results section on page 6 of the revised manuscript, where the identification of EMC-like pathway proteins is first mentioned. The hypothesized importance to matlystatin biosynthesis is also described here. This feels like a suitable location to introduce this level of detail.

3. Line 116, the compounds that are produced are described as the deshydroxy amide derivatives, which is confusing. The compounds could be better described as either deshydroxy-matlystatins or the terminal amide derivatives.

We thank the reviewer for the suggestion. In the revised manuscript the text has been changed to “deshydroxymatlystatins”

4. Line 134, not really sure what is meant by mutant strains. Is mutant strain referring to the strain with the inserted fosmid containing the unknown biosynthetic gene cluster?

We want to apologize for the confusion! Here the term “mutant” refers to the mutations that we have introduced prior to the analysis, i.e. the chromosomal integration of an exogenous gene cluster. For clarity we have modified the paragraph in the revised manuscript:

“The accumulation of deshydroxy derivatives of matlystatin A (1a) and B (2a) was observed for all heterologous strains carrying the cluster (Figure 3B). In addition, we found low amounts of

deshydroxymatlystatins D/F only in extracts of the *S. coelicolor* M1154 derivatives with the cluster (Figure S5). For the following studies we thus used *S. coelicolor* M1154 as a host.”

Reviewer 3:

“Interestingly, the authors employed a remarkable panel of complementary approaches/ tools, including advanced experiments (feeding, knockout) and manual annotation (MS/MS fragmentation) along with bioinformatics resources (AntiSmash/Blast, GNPS), which allowed them to characterize the biosynthetic assembly lines in metalloproteinase inhibitors. It really represents a state of art example for the elucidation of uncharacterized BCG.”

We would like to thank the reviewer for the very encouraging comments!

1. *In SI, material and methods: provide detailed information about source parameters for the two mass spectrometers, and their data dependent methods properties.*

In the Experimental Procedures section of the revised manuscript we have now included extensive information on the source parameters of the different mass spectrometers that were used in our study. In addition, parameters for data dependent MS experiments have been defined.

2. *Mass spectrometric data: LC-MS(-MS) data must be uploaded to a public repository such as Metabolights (<http://www.ebi.ac.uk/metabolights/>) or MassIVE (<http://massive.ucsd.edu>). Indicate that in the manuscript, and provide the deposit number.*

As suggested by the reviewer data from LC-MS measurements relevant to our study has been deposited at the public repository MassIVE. The deposit numbers have been included in the Experimental Procedures section of the revised Supporting Information and the Data Availability section.

3. *Molecular networking: Important to provide in the manuscript or SI the url link to the GNPS molecular networking job. Provide the Cytoscape file as supporting information.*

As suggested by the reviewer the link to the GNPS job has now been included in the revised manuscript in the Data Availability section and the SI. In addition, the cytoscape file that informs the network shown in Figure 5A has been uploaded as SI (Matlystatin_network.cys). An additional supplementary figure (Figure S16) displaying the total network has been added to provide context to how the data was processed to generate Figure 5A.

4. *Mass spectrometry: To promote data sharing with the community authors must deposit the MS/MS spectra of matlystatins and congeners in public spectral library. This can be achieved easily using the GNPS molecular networking job (click on View Cluster Spectra, and annotate to spectral library) In addition, this will allow the researchers to know automatically if there is any other public MassIVE dataset that would contain these compounds or related ones. Indicate that in the manuscript, and provide the deposit numbers.*

We would like to thank the reviewer for the suggestions and agree that it could be very useful for the wider natural products community to make our data accessible. The MS2 spectra of the matlystatins congeners have now been added to the public spectral library. Currently, it appears that there are no other datasets that contained compounds similar to the matlystatins (also see comments below relating to Dereplicator), although this could change when the next round of Dataset Continuous Identification analyses take place on the MassIVE datasets.

5. *Material and methods about MS: please provide more detailed information about source parameters, and data dependent methods properties.*

In the Experimental Procedures section of the revised manuscript we have now included extensive information on the source parameters of the different mass spectrometers that were used in our study. In addition, method properties for data dependent experiments have been defined where appropriate.

6. *Abstract: Articulation between sentences 1 and 2 is not obvious. It would be beneficial to present that actinonin is a metalloproteinase inhibitor.*

We thank the reviewer for the suggestion. In the abstract of the revised manuscript the second sentence has been modified to:

“Actinonin, a potent metalloproteinase inhibitor and a lead compound for the development of antibiotics, comprises a rare *N*-hydroxy-2-pentyl-succinamic acid warhead.”

7. *Line 42. Insert a reference for natural products as drug. Such as <http://pubs.acs.org/doi/abs/10.1021/acs.jnatprod.5b01055>*

The suggested reference has been included in the revised manuscript.

8. *Line 117: explicit MS2*

This has been modified according to the suggestion of the reviewer.

9. *Line 220: Instead of "indicates", I would recommend "suggests" as 4 peaks are actually observable in the figure S4.*

This has been changed according to the recommendations of the reviewer.

10. *Use MS/MS or MS2 throughout the text.*

The term MS2 is now used throughout our revised manuscript.

11. *Suggestion: If possible, I would suggest to run a dereplicator job and annotate detected compounds in GNPS spectral library.*

We thank the reviewer for the interesting suggestion. The GNPS network that is used in this manuscript (generated on 4th Jan 2017) specified that all GNPS spectral libraries were searched for matching compounds, but this yielded no known compounds associated with the matlystatin network. We have now subjected our dataset to Dereplicator analysis (parameters: search for analogs within 150 Da using Varquest with only 4 matched peaks, search the extended PNP database with a minimum of two amino acids, and additionally search for Na adducts). This did not identify any known compounds in the matlystatin network. It did provide possible hits to other peptides elsewhere in the spectra, although the annotated hits would need significant further investigation before they could confidently assigned.

General:

1. To discuss recent publications related to the biosynthesis of matlystatins and actinonin we have, in addition to the modifications described above, included a small paragraph in the first Results chapter of the revised manuscript:

“MatD and MatF have 48% and 49% sequence identity to the ornithine N5-hydroxylase KtzI and the recently characterized heme-dependent enzyme KtzT from *Kutzneria* sp. 744, respectively. This enzyme pair has been shown to be responsible for the formation of the N-N bond of Pip via the generation of an N-hydroxylated intermediate.^{16,17}”

2. Following a potential acceptance of the paper, we will submit data relating to the actinonin and matlystatin gene clusters to the Minimum Information about a Biosynthetic Gene cluster (MIBiG) server to further enable public access to our work.

Reviewers' Comments:

Reviewer #1 (Remarks to the Author):

The revision has addressed most of the reviewers' comments. Only one additional minor suggestion: The titers of compounds from gene cluster heterologous expression could also be reported to demonstrate the efficiency of these heterologous hosts.

Reviewer #2 (Remarks to the Author):

This manuscript is a resubmission that is considerably improved from the first version. The authors did an excellent job at responding to the suggestions of the first round of reviewers. Despite the concern that the recombinant enzymes have not been characterized (referee 1), the panel of complementary approaches provides strong evidence to support the conclusions, that are generally speaking, of high impact and will be of considerable interest to the scientific community.

Only one comment: Figure 2 (and other figures with structures). Noticed that some structures show charge while others do not. For example, the MatG substrate in Figure 2 shows the carboxylate, while matylstatin A and precursors show carboxylic acid. It is recommended to be consistent with the structures.

Reviewer #3 (Remarks to the Author):

The manuscript is now meeting the higher standard regarding data reproducibility for LC-MS/MS. I would simply suggest to modify figure S16 and display the m/z in the node.

Responses to Reviewers:

Reviewer #1 (Remarks to the Author): The revision has addressed most of the reviewers' comments. Only one additional minor suggestion: The titers of compounds from gene cluster heterologous expression could also be reported to demonstrate the efficiency of these heterologous hosts.

We thank the reviewer for the suggestion. We have isolated deshydroxymatlystatin A from the heterologous producer and included the following sentence in the revised manuscript: "We were able to recover deshydroxymatlystatin A from culture extracts of the heterologous producer in yields of 16 mg/L."

Reviewer #2 (Remarks to the Author): This manuscript is a resubmission that is considerably improved from the first version. The authors did an excellent job at responding to the suggestions of the first round of reviewers. Despite the concern that the recombinant enzymes have not been characterized (referee 1), the panel of complementary approaches provides strong evidence to support the conclusions, that are generally speaking, of high impact and will be of considerable interest to the scientific community. Only one comment: Figure 2 (and other figures with structures). Noticed that some structures show charge while others do not. For example, the MatG substrate in Figure 2 shows the carboxylate, while matylstatin A and precursors show carboxylic acid. It is recommended to be consistent with the structures.

We thank the reviewer for the encouraging comments. We have modified the structures in the revised manuscript according to the reviewers' suggestions.

Reviewer #3 (Remarks to the Author): The manuscript is now meeting the higher standard regarding data reproducibility for LC-MS/MS. I would simply suggest to modify figure S16 and display the m/z in the node.

Supplementary Figure 16 has been modified according to the reviewers' suggestion.